# EARLY LAYER READOUTS FOR ROBUST KNOWLEDGE DISTILLATION

## ABSTRACT

Domain generalization (DG) aims to learn a model that can generalize to unseen i.e. out-of-distribution (OOD) test domain. While large-capacity networks trained with sophisticated DG algorithms tend to achieve high robustness, they tend to be impractical in deployment. Typically, Knowledge distillation (KD) can alleviate this via an efficient transfer of knowledge from a robust teacher to a smaller student network. Throughout our experiments, we find that vanilla KD already provides strong OOD performance, often outperforming standalone DG algorithms. Motivated by this observation, we propose an adaptive distillation strategy that utilizes early layer predictions and uncertainty measures to learn a meta network that effectively rebalances supervised and distillation losses as per sample difficulty. Our method adds no inference overhead and consistently outperforms canonical ERM, vanilla KD, and competing DG algorithms across OOD generalization benchmarks.

## 1 INTRODUCTION

Deploying machine learning models in real-world scenarios requires robustness to distribution shifts (Koh et al., 2021; Huang et al., 2021), often referred to as domain generalization (DG) (Zhao et al., 2020; Robey et al., 2021) or out-of-distribution (OOD) generalization (Wald et al., 2021; Montasser et al., 2024). While high-capacity models trained with specialized DG algorithms (Gulrajani & Lopez-Paz, 2020) can achieve strong robustness, they are often prohibitively expensive in terms of computation and memory, making them impractical for deployment in resource-constrained environments.

Knowledge distillation (KD) (Hinton et al., 2015a;b; Lopes et al., 2017), has emerged as a standard approach for improving efficiency by transferring knowledge from a large teacher model to a compact student model. Beyond efficiency aspect of KD, it has recently been explored for improving OOD robustness, where vanilla KD, as shown in (Zhou et al., 2022; 2023; Huang et al., 2023) tends to yield better OOD performance compared to models trained solely with DG algorithms. However, there is still room for improvement, as KD typically treats all samples uniformly and often overrelies on the teacher's dark knowledge, making it prone to teacher-specific biases. Moreover, existing works that adapt KD for domain generalization primarily focus on using adversarially trained teachers (Nasery et al., 2022), multimodal teacher networks for additional supervision Huang et al. (2023) or ensemble of domain-specific teachers Zhao et al. (2025), leaving the role of the student network underexplored.

To address these limitations, we propose an adaptive distillation framework that modulates distillation loss based on sample difficulty via with a lightweight *forecaster* meta-network. The forecaster leverages early layer representations and uncertainty measures to estimate sample difficulty and dynamically reweight the distillation loss. This enables the student to selectively trust the teacher where appropriate while emphasizing supervision from ground-truth labels for harder or biased samples. Crucially, our design introduces no additional inference-time overhead as the forecaster is discarded post-training, making it well-suited for practical deployment.

Our work makes the following contributions:

- We identify the limitations, and opportunities for improvement in standard KD under domain shifts, noting that uniform treatment of samples and blind reliance on the teacher hinder OOD robustness.

- We propose an adaptive distillation framework with a *forecaster* meta-network that leverages early readouts to dynamically assign instance-specific weights in the loss function based on sample difficulty.

- We show that our approach improves student robustness with affecting deployment efficiency: it adds no inference-time overhead while consistently improving OOD generalization across multiple benchmarks.

## 2 BACKGROUND AND RELATED WORK

**Instance-Specific Learning.** Prior works have explore instance-specific learning to improve neural network training under noisy conditions, via instance-specific parameters such as temperature, smoothing factors, or weights. Saxena et al. (2019); Wang et al. (2018); Algan & Ulusoy (2021) introduce learnable sample and class weights to control the importance of each sample in the learning process depending on sample reliability and label noise. Ren et al. (2019); Shu et al. (2019); Raghu et al. (2021); Jain et al. (2024) adopt meta-learning (Finn et al., 2017) using small unbiased meta-samples to learn weighing functions to obtain instance-specific weights to address class imbalance, label noise and robustness. In the context of knowledge distillation, Zhao et al. (2021) propose a curriculum-based distillation with instance-level sequence learning. Iliopoulos et al. (2022) presents reweighing strategy for the student loss in knowledge distillation with unlabeled data, to eliminate potential biases from the teacher network. Sivasubramanian et al. (2022) present a bi-level objective to learn instance-specific combination of teacher-matching and supervised objectives to learn student models that are more accurate. *In this work, we introduce a meta-network which guides the student via instance-specific weighing in the KD objective. Further, we interleave the training of the meta-network with the student, without requiring complex meta-learning.*

**Early Readouts.** Prior works on Early Readouts majorly focus on early-exiting (Han et al., 2021; Xu & McAuley, 2023; Laskaridis et al., 2021; Matsubara et al., 2022) with the aim to reduce inference cost, allowing samples to "exit" at intermediate layers via auxiliary classifiers. These include dynamic early-exiting and static early-exiting mechanisms. Dynamic methods focus on balancing trade-off between speed and accuracy at inference, by early-exit mechanisms based on dynamics of internal classifiers, such as calibrated prediction confidence and entropy(Xin et al., 2020; Liu et al., 2020; Schwartz et al., 2020; Zhou et al., 2020), class mean of sample predictions (Görmez et al., 2022). In contrast to dynamic mechanisms, Sun et al. (2022) propose a hash-based early exiting for sequence learning tasks, where tokens are assigned to fixed exiting layers using a hash function. In the context of KD, Tiwari et al. (2023) use early readout errors to detect spurious feature reliance, and propose a weighing scheme to reweigh the distillation loss to reduce feature-specific bias. *In this work, we re-purpose early readouts not for exiting, but as signals to guide our meta-network, forecaster, in assigning instance-specific weights in KD for OOD robustness, while avoiding the need for handcrafted weighing functions.*

**Distillation-based Domain Generalization.** Distillation has shown promise in OOD generalization by allowing knowledge transfer from a robust teacher network, as opposed to training student network solely on a DG Algorithm (Wang et al., 2021; Huang et al., 2023). However, prior works on KD for Domain Generalization focus on teacher network or the teacher-student interaction. Wang et al. (2021) propose gradient based regularization to lower the mapping difficulty from the teacher to the student. Nasery et al. (2022) utilize adversarially fine-tuned teacher networks to improve knowledge transfer to student for OOD generalization. Huang et al. (2023) leverage CLIP teacher model along with a proposed text-based regularization scheme to enable better transfer from teacher. Zhao et al. (2025) leverage domain-specific teachers to improve student generalizability in an online KD setting. *In contrast to prior works, our method explores student-centric adaptation, leveraging early layer prediction confidences to navigate the distillation process.*

## 3 NOTATION AND PROBLEM SETUP

**Notations.** The first set of n natural numbers $\{1, 2, \ldots, n\}$ is denoted by $[n]$. The $n$-dimensional real vector space is denoted by $\mathbb{R}^n$. Vectors are typeset in lowercase bold (e.g., $\mathbf{x}$); matrices are in uppercase bold (e.g., $\mathbf{X}$); and elements are referenced by subscripts (e.g., $\boldsymbol{x}_i$, $\boldsymbol{X}_{ij}$). When needed for clarity, elements will be referenced by subscripts on square brackets (e.g., $[\boldsymbol{x}_1]_i$, $[\boldsymbol{X}_2]_{ij}$). We denote the sigmoid function by $\sigma(x) = (1 + e^{-x})^{-1}$.

**Knowledge Distillation.** Knowledge distillation (Hinton et al., 2015b) provides an efficient framework for training compact models without the need to optimize over large-scale networks, thereby reducing both memory footprint and computational overhead. In this paradigm, the compact model (student) is trained to align with both the ground-truth labels and the predictive behavior of a larger reference model (teacher). Formally, the training objective for the student consists of two components:

1. **Supervised Loss** $\mathcal{L}_{CE}$, which measures the discrepancy between the student's predictions and the ground-truth labels using cross-entropy.

2. **Distillation Loss** $\mathcal{L}_{KD}$, defined as the Kullback-Leibler divergence between the student's and teacher's output distributions. This term encourages the student to replicate the softened predictive probabilities of the teacher, which encode richer information than hard labels alone and thus facilitate more effective knowledge transfer.

**Problem Setting.** Consider a standard supervised multi-classification setting with inputs $\mathbf{x} \in \mathcal{X}$, outputs $y \in \mathcal{Y}$, and the training data $\mathcal{D} = \{(\mathbf{x}_i, y_i)\}_{i=1}^N$ where $y_i \in [K]$, $K$ denoting the total number of classes and $\mathbf{x}_i \in \mathbb{R}^d$ denoting the $i$-th input feature. Suppose, we train a student network $\mathcal{M}_{\boldsymbol{\theta}^{(S)}}$ parameterized by $\boldsymbol{\theta}^{(S)}$ and let the corresponding teacher network $\mathcal{M}_{\boldsymbol{\theta}^{(T)}}$ parameterized by $\boldsymbol{\theta}^{(T)}$ be used for teacher-student distillation. Let the student network consists of $L$ layers, where the output representation for the $\ell$-th layer $\ell \in L$ is given by $\mathbf{z}_{\boldsymbol{\theta}^{(S)}}^\ell = h_{\boldsymbol{\theta}^{(S)}}^\ell(\mathbf{x}_i) \in \mathbb{R}^{d_\ell}$ where $d_\ell$ denotes the $\ell$-th layer dimension. The final layer output logits from student network corresponding to an input $\mathbf{x}_i$ is denoted by $z_{\boldsymbol{\theta}^{(S)}}^L := \mathcal{M}_{\boldsymbol{\theta}^{(S)}}(\mathbf{x}_i)$.

**Loss Formulation.** In KD, the student network $\mathcal{M}_{\boldsymbol{\theta}^{(S)}}$ is trained by minimizing the standard cross entropy loss between the model predictions and ground truth over the training data $\mathcal{D}$ as per (1), while the teacher predictions are used to minimize the knowledge distillation term (2), defined as the KL divergence between teacher and student logits $\mathcal{M}_{\boldsymbol{\theta}^{(T)}}(\mathbf{x}_i)$ and $\mathcal{M}_{\boldsymbol{\theta}^{(S)}}(\mathbf{x}_i)$.

$$\mathcal{L}_{\text{CE}}^{(S)} = -\frac{1}{N} \sum_{i=1}^N \sum_{j=1}^K \mathbf{1}_{(y_i=j)} \log p_{\boldsymbol{\theta}^{(S)}}(\mathbf{x}_i)_j \quad (1) \quad \mathcal{L}_{\text{KD}} = -\tau^2 \frac{1}{N} \sum_{i=1}^N \mathbb{D}_{\text{KL}}\big(p_{\boldsymbol{\theta}^{(T)}}(\mathbf{x}_i) \,\|\, p_{\boldsymbol{\theta}^{(S)}}(\mathbf{x}_i)\big),$$
$$(2)$$

where $p_{\boldsymbol{\theta}}(\mathbf{x}_i)_j = \frac{\exp(\mathcal{M}_{\boldsymbol{\theta}}(\mathbf{x}_i)_j/\tau)}{\sum_{k=1}^K \exp(\mathcal{M}_{\boldsymbol{\theta}}(\mathbf{x}_i)_k/\tau)}$ denotes $j$-th class probability where $j \in [K]$. Here, $\tau$ denotes the temperature used to soften the output probabilities of both the student and the teacher. Generally, a higher temperature allows for smoother distributions, which capture richer inter-class relationship from the teacher (Hinton et al., 2015a). This allows the student to capture more nuanced relationships between classes rather than focusing on a single class representing the highest probability.

**Student Training Objective.** The student network's loss is denoted by $\mathcal{L}_{\text{tot}}^{(S)}(\cdot, \cdot; \boldsymbol{\theta}^{(S)})$, where

$$\mathcal{L}_{\text{tot}}^{(S)}(\mathcal{X}, \mathcal{Y}; \boldsymbol{\theta}^{(S)}) = \alpha \cdot \mathcal{L}_{CE}^{(S)} + \beta \cdot \mathcal{L}_{KD} \tag{3}$$

Here, $\mathcal{L}_{\text{tot}}^{(S)}(\cdot, \cdot; \boldsymbol{\theta}^{(S)})$ is expressed as a convex combination of $\mathcal{L}_{CE}^{(S)}$ and $\mathcal{L}_{KD}$, with $\alpha \in \mathbb{R}^+$ and $\beta \in \mathbb{R}^+$ being seen as two degrees of freedom, controlling the contribution of each loss component (Srivastava et al., 2015). The two degrees of freedom can be reduced to a single degree of freedom by bounding $\alpha$ and $\beta$ with the condition: $\alpha + \beta = 1$.

Moreover, in the absence of labeled data to train the student model, the distillation reduces to a soft-distillation setting, where $\alpha = 0$. In this case, the soften teacher probabilities form the only source of supervision for the student model (Lopes et al., 2017).

**Student capacity limits and Teacher Confidence.** The student's ability to learn the teacher's deeper representational space is inherently bounded, and the student may fail to capture richer information beyond a threshold due to limited capacity. On the other hand, larger teachers are over-confident, yielding higher target logits and lower variance in predictions, resulting in less distinctive incorrect-class softmax probabilities. In this case, if distillation temperature is increased, teacher guidance becomes weaker, smoothing strengthens due to the softened distribution, and class discriminability measured by variance of incorrect-class probability initially rises then falls off.

Together, both the student capacity and teacher confidence restrict the effectiveness of knowledge transfer (Li et al., 2022).

## 3.1 OOD Generalization in Distilled Models

**Bottlenecks with Vanilla KD.** While distillation serves as a good approach for model compression, the student model can be interpreted as a small clone of the teacher over-fitted to the teacher's learned patterns. The over-fitting and the student model's limited capacity typically fails to generalize well with OOD samples at inference time (Yue et al., 2023). While there exist OOD generalization algorithms targeted towards OOD robustness in neural network (Gulrajani & Lopez-Paz, 2020), the lack of representation capacity affects the ability of the student to directly benefit from these algorithms.

A general solution to improve generalization ability of the student is to train the teacher network with domain generalization (Gulrajani & Lopez-Paz, 2020) algorithms or adversarial training (**?**) (Nasery et al., 2022) for OOD robustness. During distillation, the robust teacher can act as a "shortcut" that helps the student model bypass the requirement of complex, robust training. The student model inherits the teacher's ability to produce well-calibrated outputs for seen and unseen input instances. This helps the student to be robust to distribution shifts in a deployment environment.

## 3.2 Early Readouts

Early-layer confidences in a neural network offer valuable insights about an input sample (Baldock et al., 2021). Low confidences at these layers can suggest that the sample is complex or ambiguous, potentially making it harder for the model to classify accurately. Conversely, higher early-layer confidences indicate that the low-level representations align more closely with a particular class, increasing the likelihood that the student network will classify it correctly. By providing additional supervision, such as hard labels, for ambiguous samples, the model can improve its learning of overlapping or confusing features across classes, ultimately enhancing overall performance. (Tiwari et al., 2024) utilize early exit information from neural networks.

## 4 Proposed Methodology

Our overall architecture consists of the teacher-student setup, with alterations to the student network. We incorporate auxillary networks (internal classifiers) on intermediate layers' output representations, say denoted as $\mathcal{E} \subseteq [L]$, where $L$ denotes the number of layers in the model.

Figure 1 provides a brief overview of our approach where we incorporate early-layer predictions and uncertainty measures from the student model to dynamically weigh the individual loss components of the distillation objective. In the sections to follow, we elaborate on the auxiliary networks and the *forecaster* in detail.

## 4.1 Auxiliary Network: Early Layer Confidence

**Design of Auxiliary Networks.** For each early layer $\ell \in \mathcal{E}$, we instantiate an auxiliary predictor $\mathbf{A}_{\text{aux}}^{(\ell)}(\bullet\,; \boldsymbol{\varphi}_\ell) : \mathbb{R}^{d_\ell} \to \Delta^K$, with the $\ell$-th auxiliary classifier parameterized by $\boldsymbol{\varphi}_\ell$. The input to $\mathbf{A}_{\text{aux}}^{(\ell)}$ is the $\ell$-th layer's output representation $\mathbf{z}_{\boldsymbol{\theta}(S)}^\ell$. The role of the auxiliary classifier is to encode how well the truncated feature stack up to layer $\ell$ differentiates among the label space $\mathcal{Y}$, i.e., the extent to which intermediate layers capture discriminative information over class labels. Such auxiliary predictors have been shown to be effective both for improving gradient flow and feature discriminability during training (Szegedy et al., 2015; Lee et al., 2015), as well as for post-hoc analysis of representation quality via probing (Alain & Bengio, 2016).

**Training Objective of Auxiliary Networks.** On the training split $\mathcal{D}$, the objective of each auxiliary encoder is to minimize the standard classification loss:

$$\mathcal{L}_{\text{aux}}^{(\ell)}(\mathbf{z}_{\boldsymbol{\theta}(S)}^\ell\,; \boldsymbol{\varphi}_\ell) = -\sum_{i=1}^N \sum_{j=1}^K \mathbf{1}_{\{y_i=j\}} \log \frac{\exp\big(\mathbf{A}_{\text{aux}}^{(\ell)}(\mathbf{z}_{\boldsymbol{\theta}(S)}^\ell\,; \boldsymbol{\varphi}_\ell)_j\big)}{\sum_{k=1}^K \exp\big(\mathbf{A}_{\text{aux}}^{(\ell)}(\mathbf{z}_{\boldsymbol{\theta}(S)}^\ell\,; \boldsymbol{\varphi}_\ell)_k\big)}. \tag{4}$$

## 4.2 Forecaster Meta-Network

Deep neural networks are prone to overfitting under label noise or class imbalance, where fixed re-weighting schemes often fail due to their reliance on handcrafted weighting functions and hyper-parameters. To address this, we introduce a *forecaster* module that adaptively maps sample-level

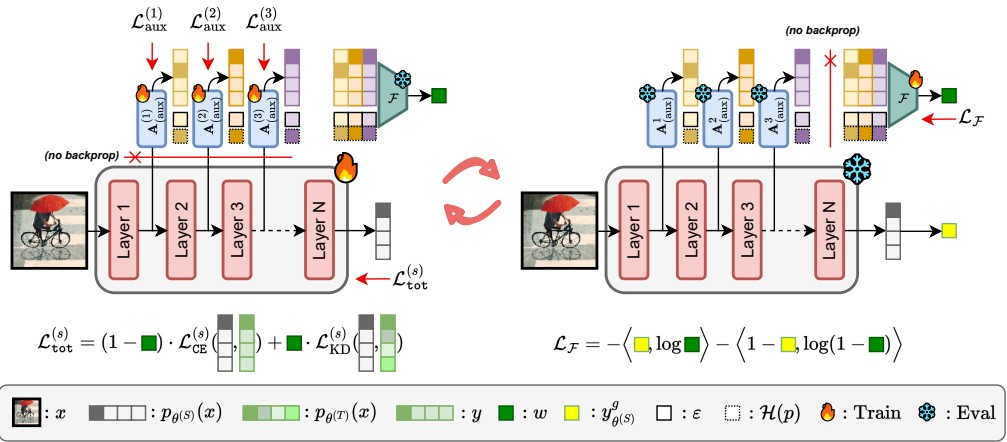

Figure 1: Overview of adaptive knowledge distillation that leverages early layer readout predictions and uncertainty measures to learn a meta-network *forecaster* which modulates contribution of ground-truth supervision and teacher supervision in the distillation process at an instance level. Here, the student network, early layer auxiliary networks and *forecaster* are trained in an interleaved fashion. First, the student backbone and the auxiliary network are trained on the classification objective, while the *forecaster* remains frozen, for a fixed number of train minibatches. Next, for a fixed number meta-validation set from the training domains is used to train the forecaster with the objective to estimate student correctness. At test time, the *forecaster* and auxiliary networks are discarded, resulting in our method requiring the same computational resources as a vanilla KD during inference.

statistics to re-weigh the influence of the teacher network in the distillation process. Parameterized as a lightweight neural network and optimized jointly with the model, the forecaster provides a flexible data-driven mechanism that generalizes beyond handcrafted weighting rules. We denote the forecaster as $\mathcal{F}(\bullet; \psi^f)$ defined as a meta-network parameterized by $\psi^f$.

**Confidence Margin** The layer wise confidence margin of the $\ell$-th auxiliary network denoted by $\varepsilon^{(\ell)}$ acts a good signal of uncertainty in the predictions at the early layers (Tiwari et al., 2024; Xin et al., 2020; Liu et al., 2020) where $p_{\max}(\varphi_l) = \max(p_1(\varphi_l), p_2(\varphi_l), \dots, p_K(\varphi_l))$ and $p_j(\cdot)$ indicates the probability of the auxiliary layer $l$'s output being classified as the $j$-th label, $\forall j \in [K]$.

$$\varepsilon^{(\ell)} = p_{\max}(\cdot) - \max(p_i(\cdot) \mid i \neq \arg\max(p_i(\cdot))) \quad (5)$$

Particularly, *forecaster* is fed logits ($\mathbf{z}$), Entropy $\{\mathcal{H}(p)\}$ and Confidence Margin ($\varepsilon$) of the auxiliary network predictions as uncertainty indicators of auxiliary network, highlighting prediction confidence and randomness.

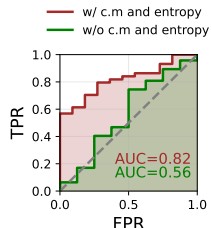

Figure 2: ROC curves showcasing the impact of incorporating confidence margin ($\varepsilon$) and entropy $\mathcal{H}(p)$ features from early layers into the forecaster.

**Training Forecaster** $\mathcal{F}(\bullet; \psi^f)$ The Forecaster is trained in conjunction to the student $\mathcal{M}_{\theta^{(S)}}$ and the auxiliary networks $\mathbf{A}_{\text{aux}}^{(\ell)}(\bullet; \varphi)$ using a binary classification objective. The binary classification task for the forecaster is to determine whether student model correctly predicts an input instance. Higher output probability from the forecaster indicates student is more likely to correctly classify the sample, indicating an easier sample. We utilize the output probability of the forecaster directly to weigh the KL divergence between the student and the teacher, forcing the student to mimic teacher for such samples and focus more on ground-truth supervision when the forecaster output probability is smaller. To train the forecaster, we minimize $\mathcal{L}_{\text{focs}}(\theta^{(S)}; \psi^f)$ on the validation split $\mathcal{D}_v \subseteq \mathcal{D}$

$$\mathcal{L}_{\text{focs}}(\theta^{(S)}, \psi_f) = -\left\langle \mathbf{y}_{\theta^{(S)}}^g, \log \mathbf{w}_{\psi_f} \right\rangle - \left\langle \mathbf{1} - \mathbf{y}_{\theta^{(S)}}^g, \log(1 - \mathbf{w}_{\psi_f}) \right\rangle, \quad (6)$$

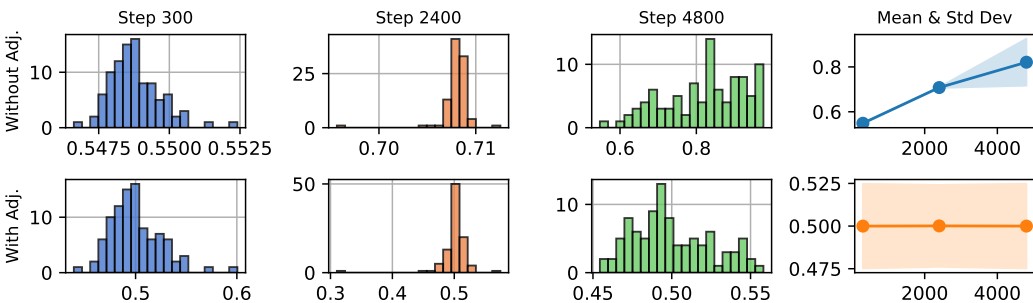

Figure 3: Distribution of forecaster outputs at different training steps, comparing the unadjusted case (**top row**) with post-hoc adjustment (**bottom row**). The rightmost plots summarize the deviations across epochs. Post-hoc adjustment stabilizes the output distribution, negating drift towards skewed values.

where $\mathbf{y}^g_{\boldsymbol{\theta}(S)} \in \{0,1\}^N$ is the binary vector of ground-truth labels for the student model predictions, with entries $y^g_{\boldsymbol{\theta}(S),i} = \mathbf{1}_{(\hat{y}_i = y_i)}$ for each instance $i \in [N]$, and $\mathbf{w}_{\boldsymbol{\psi}_f} \in (0,1)^N$ is the vector of forecaster outputs parameterized by $\psi_f$. In a mini-batch setting (say batch size $B$), the forecaster loss is averaged across all samples in the batch with $\mathbf{w}_{\boldsymbol{\psi}_f} \in (0,1)^{|B|}$.

**Design of Forecaster.** The *forecaster* is implemented as a lightweight neural network comprising a 1D convolution on stacked intermediate logits obtained from auxiliary networks. The convolution operation integrates the logits across layers, one class at a time. To this representation, we append uncertainty measures: confidence margin and entropy of each Auxiliary Network prediction. Using a linear projection on this intermediate representation, the *forecaster* provides a scalar weight $w_{(\boldsymbol{\psi}^f)} \in [0,1]$ for each instance in the minibatch, which indicates the confidence of the *forecaster* in assessing whether the student network will correctly predict the input instance.

**Post-hoc Adjustment of Forecaster Output.** The forecaster $\mathcal{F}(\boldsymbol{z}_f; \boldsymbol{\psi}^f)$ outputs a weight $w_t \in [0,1]$. For a batch of input instances, forecaster output distribution can be skewed, affecting the stability of student model training and individual loss component contributions. Therefore, for a batch of samples $(B)$, we employ the following post-hoc adjustment on the forecaster output logits $(\boldsymbol{z}_f)$ before utilizing the output for training the student, $\mathbf{w}^{\text{adj}}_{\boldsymbol{\psi}_f} = \sigma\left(\varsigma^*\left(\frac{\boldsymbol{z}_f - \mu_B}{\varsigma_B}\right) + \mu^*\right)$, where $\mu^*$ and $\varsigma^*$ are hyperparameters. Ultimately, the student is trained by minimiz-

| METHOD | A | C | P | R | AVG. |
|---|---|---|---|---|---|
| KD (Hinton et al., 2015b) | 52.7 | 49.7 | 72.1 | 74.2 | 62.2 |
| KD+F | **54.3** | 50.2 | 70.5 | 72.5 | 61.9 |
| KD+F+ADJ. | 54.0 | **51.3** | **71.6** | **75.1** | **63.0** |

Table 1: **OOD classification accuracy** (%) **on the OfficeHome dataset**. KD denotes vanilla knowledge distillation, F incorporates the forecaster, and ADJ. further applies the proposed post-hoc adjustment to forecaster outputs. The adjustment ($\mu^* = 0.5$ and $\varsigma^* = 0.1$) consistently improves average performance across held-out domains.

ing, $\mathcal{L}^{(s)}_{\text{tot}}(\boldsymbol{\mathcal{X}}_B, \mathcal{Y}_B; \boldsymbol{\theta}^{(S)}, \mathbf{w}_{\boldsymbol{\psi}_f}) = \frac{1}{|B|} \sum_{j=1}^{|B|} (1 - \mathbf{w}^{\text{adj}}_{\boldsymbol{\psi}_f}[j]) \cdot \ell^{(s)}_{\text{CE}}[j] + \mathbf{w}^{\text{adj}}_{\boldsymbol{\psi}_f}[j] \cdot \ell^{(s)}_{\text{KD}}[j]$.

## 5 BI-LEVEL OPTIMIZATION PROBLEM AND CONVERGENCE

The interleaved training schedule between the student network and the forecaster can be viewed as a bilevel optimization problem given by,

$$\arg\min_{\mathbf{w}_f} \mathcal{L}_{\text{focs}}\Big( \overbrace{\underbrace{\arg\min_{\boldsymbol{\theta}} \mathcal{L}^{(s)}_{\text{tot}}(\boldsymbol{\theta}, \mathbf{w}_f)}_{\text{inner-level}}, \mathcal{D}_v}^{\text{outer-level}} \Big)$$

Starting at $\boldsymbol{\theta}^{(S)}_0$ and $\mathbf{w}^0_f$, a single gradient update can be written as,

$$\boldsymbol{\theta}_t^{(S)} \leftarrow \boldsymbol{\theta}_{t-1}^{(S)} - \eta_1 \nabla_{\boldsymbol{\theta}_{t-1}^{(S)}} \mathcal{L}_{\text{tot}}^{(s)}(\cdot; \boldsymbol{\theta}_{t-1}^{(S)}, \mathbf{w}_f^t) \tag{7}$$

$$\text{where } \mathbf{w}_f^t \leftarrow \mathbf{w}_f^{t-1} - \eta_2 \nabla_{\mathbf{w}_f^{t-1}} \mathcal{L}_{\text{focs}}(\boldsymbol{\theta}_{t-1}^{(S)}, \mathbf{w}_f^{t-1}) \tag{8}$$

## 6 ALGORITHM

We outline the distillation process with an interleaved training process where the student model, auxiliary networks and the forecaster are trained in an alternating pattern. For pre-determined $k$ steps, the primary student model and the auxiliary networks are trained on the training domains using the training split of the training domains. Here, the forecaster remains frozen, and is used to generate instance-level weights to guide the student training. Subsequently, for next $k$ steps, the forecaster is trained on the on a held-out validation split from the training data of the training domain(s) with the lastest checkpoints of the student model and the auxiliary networks, which are frozen. This process continues until the exit criterion of the algorithm is satisfied. Refer Algorithm 1 for an illustrated outline of the algorithm.

---

**Algorithm 1** Interleaved Training of Student and Forecaster

---

**Require:** Student $\mathcal{M}_{\boldsymbol{\theta}^{(S)}}$, Teacher $\mathcal{M}_{\boldsymbol{\theta}^{(T)}}$, Forecaster $\mathcal{F}(\boldsymbol{\psi}_f)$; $\mathcal{B}_{\text{train}}, \mathcal{B}_{\text{val}}$; $T_s$ (student steps), $T_f$ (forecaster steps), $N$ (total steps), $L$ (number of layers), $\mu^*$ (Adjustment mean), $\varsigma^*$ (Adjustment st. deviation)
1: Initialize $\boldsymbol{\theta}_0^{(S)}, \boldsymbol{\theta}_0^{(T)}, \boldsymbol{\psi}_0^f, [\varphi_0^{(\ell)}]_{\ell=1}^{L-1}$
2: **for** $t = 1$ to $N$ **do**
3:     **if** $t \bmod (T_s + T_f) < T_s$ **then**
4:         ▶ *Student and Auxiliary Network Update*
5:         Set $\mathcal{M}_{\boldsymbol{\theta}^{(S)}}, \mathcal{M}_{\boldsymbol{\theta}^{(T)}}, \mathcal{F}(\boldsymbol{\psi}_f), [\mathbf{A}_{\text{aux}}^{(\ell)}]_{\ell=1}^{L-1}$ to **eval**
6:         $\mathbf{X} \leftarrow \text{concat}(\mathbf{x} \mid (\mathbf{x}, \mathbf{y}) \in \mathcal{B}_{\text{train}})$
7:         $[\mathbf{A}_{\text{aux}}^{(\ell)}(\mathbf{X}; \boldsymbol{\varphi})]_{\ell=1}^{L-1}, \hat{\mathbf{Y}} \leftarrow \mathcal{M}_{\boldsymbol{\theta}^{(S)}}(\mathbf{X})$
8:         $\mathbf{O} \leftarrow \mathcal{F}([\mathbf{A}_{\text{aux}}^{(\ell)}(\mathbf{X}; \boldsymbol{\varphi})]_{\ell=1}^{L-1})$
9:         $\mu_B = \mathbf{O}.\text{mean}(), \varsigma_B = \mathbf{O}.\text{std}()$
10:        $\mathbf{W} = \sigma\left(\varsigma^*\left(\frac{\mathbf{O} - \mu_B}{\varsigma_B}\right) + \mu^*\right)$
11:        Set $\mathcal{M}_{\boldsymbol{\theta}^{(S)}}$ to **train**;
12:        $\boldsymbol{\theta}_t^{(S)} \leftarrow \boldsymbol{\theta}_{t-1}^{(S)} - \frac{\eta}{|\mathcal{B}_{\text{train}}|} \sum_{\mathbf{x} \in \mathcal{B}_{\text{train}}} \mathcal{L}_{\text{tot}}^{(s)}(\mathbf{x}; \boldsymbol{\theta}_{t-1}^{(S)}, \mathbf{W})$
13:        **for** $\ell = 1$ to $L - 1$ **do**
14:            Set $\mathbf{A}_{\text{aux}}^{(\ell)}$ to **train**;
15:            $\varphi_t^{(\ell)} = \varphi_{t-1}^{(\ell)} - \frac{\eta}{|\mathcal{B}_{\text{train}}|} \sum_{\mathbf{x} \in \mathcal{B}_{\text{train}}} \mathcal{L}_{\text{aux}}^{(\ell)}(\mathbf{A}_{\text{aux}}^{(\ell)}(\mathbf{x}); \varphi_{t-1}^{(\ell)})$
16:        **end for**
17:     **else**
18:         ▶ *Forecaster Update*
19:         Set $\mathcal{F}(\boldsymbol{\psi}_f)$ to **train**, $\mathcal{M}_{\boldsymbol{\theta}^{(S)}}, [\mathbf{A}_{\text{aux}}^{(\ell)}(\bullet; \boldsymbol{\varphi})]_{\ell=1}^{L-1}$ to **eval**
20:        $\boldsymbol{\psi}_t^f \leftarrow \boldsymbol{\psi}_{t-1}^f - \frac{\eta_f}{|\mathcal{B}_{\text{val}}|} \sum_{\mathcal{B}_{\text{val}}} \mathcal{L}^{(f)}(\boldsymbol{\theta}^{(S)}, \boldsymbol{\psi}_{t-1}^f)$
21:     **end if**
22: **end for**

---

## 7 EXPERIMENTS

**Evaluation.** For the image-classification experiments, we leverage the DomainBed Suite (Gulrajani & Lopez-Paz, 2020). DomainBed provides a fair, standardized and reproducible setup for evaluating and comparing our method against best performing subset from a wide range of DG algorithms including ERM (Vapnik, 1998), GroupDRO (Sagawa et al., 2020), Mixup (Yan et al., 2020), MLDG (Li et al., 2017b), CORAL (Sun & Saenko, 2016), MMD (Li et al., 2018a), DANN (Ganin et al., 2016) and C-DANN (Li et al., 2018b). For the text-classification experiments, we leverage representative ID-OOD task pairs from GLUE-X (Yang et al., 2023). In the case of text-classification, we compare our method against vanilla KD and ERM. We use classification accuracy as the primary metric.

**Model Pool.** In our KD experiments, we distill from a robust teacher model. We experiment with both ResNet (He et al., 2015) and Vision Transformer (Dosovitskiy et al., 2021) as the teacher network. Specifically, we first identify the best-performing domain generalization algorithm at the teacher level per dataset using `ResNet-152` and `ViT-L/16`, and use the best teacher network in the distillation process. For the student network, we select the smallest capacity model from the ResNet family: ResNet-18. For transformer-based text experiments, we leverage Encoder-only `bert-base-uncased` (Devlin et al., 2019) as the teacher network. In this case, the student network is a 4-layer compact BERT variant: `google/bert_uncased_L-4_H-256_A-4` (Turc et al., 2019).

| DATASET | DOMAINS | INSTANCES | LABELS |
|---|---|---|---|
| OFFICEHOME | 4 | 15,588 | 65 |
| PACS | 4 | 9,991 | 7 |
| VLCS | 4 | 10,729 | 5 |
| TERRAINCOGNITA | 4 | 24,778 | 10 |
| PARAPHRASE | 2 | 44,118 | 2 |
| NLI | 2 | 402,517 | 3 |

Table 2: Dataset statistics.

**Datasets.** We evaluate and compare our method on four benchmark datasets, including OfficeHome (Venkateswara et al., 2017), PACS (Li et al., 2017a), VLCS (Fang et al., 2013) and TerraIncognita (Beery et al., 2018). For text-based benchmarking, we choose the tasks of (1) paraphrase identification with MPRC as ID split and QQP as OOD split, and (2) natural language inference (NLI) with MNLI as ID split and MNLI-mismatched as OOD split. Table 2 provides the details of the domains, instances, and label counts for each dataset.

**Baselines.** We compare our method against three baselines. We include the Empirical Risk Minimization (ERM) (Vapnik, 1998) as a canonical domain generalization algorithm, supported by findings in Gulrajani & Lopez-Paz (2020). Then, we select the best-performing OOD algorithm identified at the teacher level as the second baseline. In this case, we retrain the student model without distillation. Finally, we include vanilla KD (Hinton et al., 2015b), where the student us trained to mimic the logits of the teacher without any further modifications. In the case of text-classification, we compare our method against vanilla KD and ERM.

**Training Protocol.** For a dataset comprising $N$ domains, we adopt the leave-one-domain-out setup where the network is trained on $N - 1$ domains and evaluated on the held-out domain. We reserve 20% from each of training domain as a validation set for model selection. In our method, remaining 80% training split is further divided, with 90% used to train the student backbone and auxiliary networks, and 10% held out to train the forecaster. We leverage gold validation splits to train the forecaster in our text-modality evaluations. Within the distillation setup, we choose temperature $\tau = 2$ in the KD loss formulation (Hinton et al., 2015a).

**Auxiliary Network Design.** For the ResNet student, each residual block (Total: 4) stage yields feature maps that encode progressively abstract representations of the input (He et al., 2015). To obtain class-wise representations from these intermediate features, we employ lightweight auxiliary heads following the design principles of prior works on multi-layer feature supervision (Szegedy et al., 2015; Lee et al., 2015) and early-exits (Xin et al., 2020; Liu et al., 2020). Each auxiliary head comprises two components: (i) a pooling operation to reduce the spatial dimensionality, and (ii) a feed-forward projection layer that maps the pooled features to a vector of dimension equal to the number of target classes. For transformer-based BERT backbone, the auxiliary networks are simple linear projections i.e. one layer feed-forward block.

## 8 RESULTS

Table 3 reports the domain generalization performance across four benchmark datasets. Here, ERM Vapnik (1998) corresponds to standard empirical risk minimization, where the network is trained solely on ground-truth supervision. Similarly, DB. SOTA refers to the strongest domain generalization algorithm among GroupDRO (Sagawa et al., 2020), Mixup (Yan et al., 2020), MLDG (Li et al., 2017b), CORAL (Sun & Saenko, 2016), MMD (Li et al., 2018a), DANN (Ganin et al., 2016) and C-DANN (Li et al., 2018b), selected as per findings described in Appendix D.1. In the KD (Hinton et al., 2015a) setting, the student is trained with an additional supervision from a larger network fine-tuned using the best-performing DG algorithm.

We observe that KD consistently outperforms both, the canonical empirical risk minimization (ERM) and the dataset-specific best-performing DG algorithm. This highlights the effectiveness

| METHOD | ARCH. | OFFICEHOME | PACS | VLCS | TERRA. | AVG. |
|---|---|---|---|---|---|---|
| ERM (Vapnik, 1998) | ResNet-18 | 58.4 | 79.6 | 71.6 | 43.4 | 63.3 |
| DB. SOTA | ResNet-18 | 60.2 | 80.1 | 71.9 | 42.6 | 63.7 |
| TEACHER NETWORK: ResNet-152 | | | | | | |
| KD (Hinton et al., 2015b) | ResNet-18 | 62.2 | 82.4 | 73.5 | 43.6 | 65.4 |
| KD$_{+F+ADJ.}$ (Ours) | ResNet-18 | **63.0** | **82.8** | **74.7** | **45.1** | **66.4** |
| TEACHER NETWORK: ViT-L/16 | | | | | | |
| KD (Hinton et al., 2015b) | ResNet-18 | 63.7 | **81.4** | 75.2 | 40.6 | 65.2 |
| KD$_{+F+ADJ.}$ (Ours) | ResNet-18 | 63.7 | 81.0 | **76.1** | **44.7** | **66.4** |

Table 3: Domain generalization accuracy (%) on four benchmark datasets. Best results are high-lighted in bold and green. Here, DB. SOTA refers to the best-performing DG algorithm for the dataset, selected according to the evaluation protocol described in Appendix D.1.

| METHOD | ARCH. | PARAPHRASE | NLI | Avg. |
|---|---|---|---|---|
| ERM (Vapnik, 1998) | bert_uncased_L-4_H-256_A-4 | 57.1 | 74.6 | 65.8 |
| TEACHER NETWORK: bert-base-uncased | | | | |
| KD (Hinton et al., 2015b) | bert_uncased_L-4_H-256_A-4 | 58.4 | 74.8 | 66.6 |
| KD$_{+F+ADJ.}$ (Ours) | bert_uncased_L-4_H-256_A-4 | **60.9** | **75.4** | **68.2** |

Table 4: OOD Classification accuracy (%) on (MPRC, QQP) and (MNLI, MNLI-mismatched) ID, OOD pairs from GLUE-X benchmark. Prompt details present in Appendix A.2.

of knowledge transfer from a robust teacher network. Building on the strong KD baseline, we evaluate our adaptive distillation setup, which augments KD with the forecaster meta-network that utilizes early readout signals from the student network. Our method improves over KD under both, a convolutional ResNet teacher and transformer based ViT teacher. With a robust ResNet-152 teacher, our approach achieves an average OOD accuracy of 66.4%, outperforming KD by +1.0%. Similarly, with a robust ViT-L/16 teacher, our method surpasses KD by +1.2%. These consistent improvements across datasets showcase our approach as a principled extention to the standard KD problem for domain generalization.

Table 4 reports the domain generalization results in the text modality with a transformer-based student architecture. Here, our method yields an average increase of 1.6% in accuracy, relative to the standard KD, showcasing the generality of our work across modality and model architecture.

**The Need For Post-hoc Adjustment.** As shown in Table 1, in the absence output adjustment, adaptive KD with forecaster yields lower OOD performance as compared to vanilla KD. This motivates us to understand why an unadjusted forecaster may fail. We therefore analyze the dynamics of the forecaster outputs during the distillation process with and without post-hoc adjustment to the forecaster outputs. Figure 3 showcases the change in forecaster outputs for a minibatch as training progresses. Without any correction to the forecaster output, the distribution gradually drifts towards extreme values close to 1. This drift can be attributed to the nature of the training of the forecaster. The forecaster is trained as a binary classifier to predict correctness of student network based on early readout predictions and uncertainty signals. Naturally, as the student improves, an increasing fraction of samples become easy, pushing forecaster towards overconfident predictions. This causes the weight to concentrate more to $\mathcal{L}_{KD}$ as the training progresses, making model overfit to teacher's supervisory signals, affecting its generalizability.

**The Role of Uncertainty Signals.** We also analyze the impact of early layer uncertainty features on forecaster quality. As shown in Figure 2, including entropy $\mathcal{H}(p)$ and confidence margin ($\varepsilon$) from auxiliary predictions substantially improves the forecaster's predictive ability to determine easy and hard samples, with an increase in AUC by +26%. This demonstrates that early readouts provide rich signals of sample ambiguity, which can be leveraged by the forecaster, to effectively modulate the loss weighing. Together, these emipirical ablations on the forecaster confirm that both

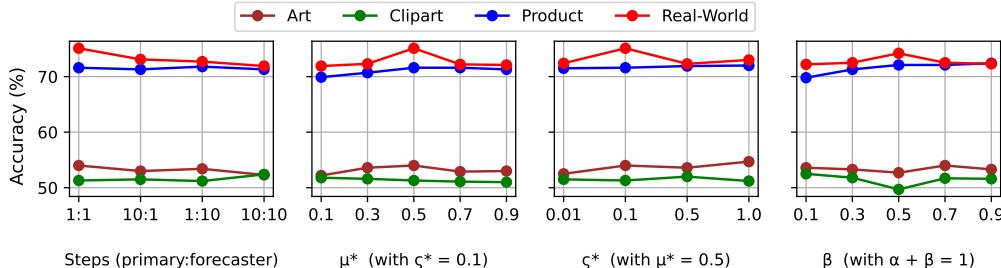

Figure 4: From left to right: Domain-wise OfficeHome OOD accuracies as a function of (a) interleaved training schedule, (b) adjusted forecaster output mean ($\mu*$), (c) adjusted forecaster output spread ($\varsigma*$) and (d) coefficient of KD loss in Vanilla KD.

post-hoc adjustment and statistical uncertainty-aware design choice is essential to the forecaster meta-network.

**Sensitivity Analysis.** We chose $\mu*$ and $\varsigma$ to be 0.5 and 0.1 as pragmatic, dataset agnostic hyperparameters. $\mu* = 0.5$ centers the post-hoc adjustment to allow equal contribution from teacher soft predictions and ground-truth supervision. This allows the training signal to not be overly dominated by the teacher or the ground-truth. $\varsigma = 0.1$ allows a controlled spread around the center, allowing an instance-based adjustment proposed by the forecaster without a strong drift in either direction. We conduct sensitivity analysis by (1) varying $\mu* \in \{0.1, 0.3, 0.5, 0.7, 0.9\}$ while keeping $\varsigma$ fixed to 0.1 and (2) varying $\varsigma \in \{0.01, 0.1, 0.5, 1.0\}$ keeping $\mu*$ fixed to 0.5. As shown in Figure 4 OOD performance does not degrade sharply across various choices of ($\mu*$, $\varsigma*$) pairs, suggesting that our method is not sensitive to the hyperparameter choices of $\mu*$ and $\varsigma*$. While the hyperparameters can be tuned per-dataset or per-domain within a dataset, we recommend the practical default choices under compute constraints.

Similarly, we set the student model update frequency, $T_s = 1$ and forecaster update frequency, $T_f = 1$ in our experiments. With $T_s = 1$ and $T_f = 1$, we simply ensure that the forecaster is updated as frequently as the student, preventing the forecaster weights becoming stale as the student representation evolves during training. To assess how the frequency of updates to the forecaster affect the downstream OOD performance of the student network, we conduct a sensitivity analysis by varying $T_s$ and $T_f$ across several configurations shown in Figure 4. As shown, we observe a small but consistent drop in average OOD performance in these alternative configurations. These observations suggest that the forecaster network is negatively affected in both cases of (1) infrequent updates ($T_s = 10, T_f = 1$) where the forecaster is lagging behind current student configuration and too frequent updates ($T_s = 10, T_f = 10$ or $T_s = 1, T_f = 10$) where the forecaster potentially overfits to current student configuration.

## 9 CONCLUSION

In this work, we introduced a *forecaster* based re-weighing approach to standard offline knowledge distillation setting, where the *forecaster* leverages early layer readouts from the student model to adaptively modulate the distillation objective, resulting in an improved generalizability of the student network as opposed to vanilla KD. Our experiments across multiple benchmarks spanning two modalities: (1) vision and (2) text, demonstrate the efficacy of our approach in improving OOD generalization of student network over vanilla KD baseline and canonical DG algorithms. Further, we provide insights on critical design choices for the *forecaster*: (1) post-hoc adjustment, which prevents *forecaster* collapse that results in overconfident predictions, and (2) Use of uncertainty measures such as entropy and confidence margin, which significantly improve *forecaster's* ability to distinguish between easy and difficult samples. Together, we establish our framework as a novel extension to standard offline KD, allowing robust generalization to unseen domains, from a student-centric design choice.

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

# Supplementary Material: Early Layer Readouts for Robust Knowledge Distillation

## CONTENTS

# A  IMPLEMENTATION AND HYPERPARAMETERS

## A.1  SEED AND HYPERPARAMETER SELECTION

For a fair comparison and reproducibility, we follow the default hyperparameter settings reported in Gulrajani & Lopez-Paz (2020) and use their released codebase as the foundation of our implementation.

## A.2  PROMPT DETAILS FOR TEXT CLASSIFICATION

```
"""Paraphrase Identification: Train Prompt (MPRC)"""

Sentence 1: {{sentence1}}
Sentence 2: {{sentence2}}
Question: Do both sentences mean the same thing?
Answer:

"""Paraphrase Identification: Test Prompt (QQP)"""

Question 1: {{question1}}
Question 2: {{question2}}
Question: Do both questions ask the same thing?
Answer:

"""Natural Language Inference: Train/Test Prompt (MNLI)"""

{{premise}}
Question: {{hypothesis}} True, False or Neither?
Answer:
```

# B  THEORETICAL RESULTS

## B.1  GENERALIZATION ERROR BOUNDS FOR OUR METHOD

We utilise the approach presented in (Ye et al., 2021) to calculate the generalization error bounds incurred via our method.

**Setup**

Let $\xi_{\text{all}}$ be the domain set we want to generalize to, and $\xi_{\text{avail}} \subseteq \xi_{\text{all}}$ be the available domain set. We denote $(X^e, Y^e)$ as the input-label pair from the data distribution of domain $e \in \xi_{\text{avail}}$.

For OOD generalization, the goal is to find a classifier $f^*$ such that it *minimizes* the *worst-domain* loss on $\xi_{\text{all}}$

$$f^* = \arg\min_{f \in \mathcal{F}} \mathcal{L}(\xi_{\text{all}}, f), \mathcal{L}(\xi, f) \equiv \max_{e \in \xi} \mathrm{E}[\ell(f(X^e), Y^e)]$$

**Definition 1** (Variation). *The variation of a feature $\phi(\cdot)$ across a domain set $\xi$ is defined as*

$$\mathcal{V}_\rho(\phi, \xi) = \max_{y \in Y} \sup_{e, e' \in \xi} \rho(P(\phi_e \mid y), P(\phi_{e'} \mid y)). \tag{9}$$

*A feature $\phi(\cdot)$ is said to be $\varepsilon$-invariant across $E$ if $\varepsilon \geq \mathcal{V}(\phi, \xi)$. where $\rho$ indicates any distribution divergence metric. (We omit the subscript $\rho$ when there is no ambiguity.)*

Let the optimal weights learnt via the forecaster be denoted as $\mathbf{w}_{\psi_f^*}$. Post adjustment, let the optimal adjusted weights be denoted as $\mathbf{w}_{\text{adj}}^*$. The optimal weights are across all samples in the batch.

$$\Re_{\text{aux}}^{(\ell)}(\boldsymbol{\theta}, \boldsymbol{\varphi}^{(\ell)}; e) := \mathbb{E}_{(X^e, Y^e)}[\mathcal{L}_{\text{aux}}^{(\ell)}(A_{\text{aux}}^{(\ell)}(h_\ell(\mathbf{X}^e; \boldsymbol{\theta}); \boldsymbol{\varphi}^{(\ell)}), Y^e)]$$

**Assumption**  There exists a strictly increasing function $\delta_l : [0, 0.5] \to [0, \infty)$ such that

$$\Re_{\text{aux}}^{(\ell)}(\boldsymbol{\theta}, \boldsymbol{\varphi}^{(\ell)}; e) - \Re_{\text{aux}}^{(\ell)}(\boldsymbol{\theta}, \boldsymbol{\varphi}_*^{(\ell)}; e) \geq \delta_l(\Delta^{(l)}(e))$$

where $\Re_{\text{aux}}^{(\ell)}(\boldsymbol{\theta}, \boldsymbol{\varphi}_*^{(l)}; e)$ indicates the Bayes risk on domain $e$ and $\Delta^{(l)}(e)$ is the average $0 - 1$ error of the auxiliary classifier at layer $l$.

**Assumption (Lipschitz in feature space)**: We assume there exists $\mathbb{C}_{\text{aux}} > 0$ such that for all $z_1, z_2$ and labels $y$:

$$|\mathcal{L}_{\text{aux}}^{(\ell)}(\mathbf{z}_1^\ell(\boldsymbol{\theta}^{(S)}); \boldsymbol{\varphi}_\ell) - \mathcal{L}_{\text{aux}}^{(\ell)}(\mathbf{z}_2^\ell(\boldsymbol{\theta}^{(S)}); \boldsymbol{\varphi}_\ell)| \leq \mathbb{C}_{\text{aux}}. \left\| \mathbf{z}_1^\ell(\boldsymbol{\theta}^{(S)}) - \mathbf{z}_2^\ell(\boldsymbol{\theta}^{(S)}) \right\|_2$$

**Domain Disagreement at layer $\ell$**

Given two distinct domains $e_1, e_2 \in \xi_{\text{avail}}$, we check how the difference in auxilliary classifier risks relates to the induced feature distribution difference.

$$\Delta\Re_{\text{aux}}^{(\ell)}(e_1, e_2) := |\Re_{\text{aux}}^{(\ell)}(\boldsymbol{\theta}, \boldsymbol{\varphi}^{(\ell)}; e_1) - \Re_{\text{aux}}^{(\ell)}(\boldsymbol{\theta}, \boldsymbol{\varphi}^{(\ell)}; e_2)|$$

Using Lipschitz assumption and total variation distance on $\mathbb{P}_e^\ell$ we have

$$\Delta\Re_{\text{aux}}^{(\ell)}(e_1, e_2) \leq \mathbb{C}_{\text{aux}} \int \left\| \boldsymbol{z} \right\|_2 |p_{e_1}^l(\boldsymbol{z}, y) - p_{e_2}^l(\boldsymbol{z}, y)| \partial \boldsymbol{z} \partial y$$

Assuming we consider a bounded feature radius at layer $\ell$: $\left\| z \right\|_2 \leq R_l$, we have

$$\Delta\Re_{\text{aux}}^{(\ell)}(e_1, e_2) \leq \mathbb{C}_{\text{aux}} R_l \int |p_{e_1}^l(\boldsymbol{z}, y) - p_{e_2}^l(\boldsymbol{z}, y)| \partial \boldsymbol{z} \partial y = 2\mathbb{C}_{\text{aux}} R_l \rho(\mathbb{P}_{e_1}^\ell, \mathbb{P}_{e_2}^\ell)$$

Thus,

$$\rho(\mathbb{P}_{e_1}^\ell, \mathbb{P}_{e_2}^\ell) \geq \frac{\Delta\Re_{\text{aux}}^{(\ell)}(e_1, e_2)}{2\mathbb{C}_{\text{aux}} R_l}$$

**Remark 1.** *The above indicates that any variation in the auxillariy risks across domains is almost surely upper bounded the TV(total variation) distance between feature label distributions for that layer l.*

Since by definition we know

$$\mathcal{V}(\beta^\top h_\ell, \xi_{\text{avail}}) := \sup_{e_1, e_2 \in \xi_{\text{avail}}} \rho(\mathbb{P}(\beta^\top h_\ell(\mathbf{X}^{e_1}) \mid Y^{e_1}), \mathbb{P}(\beta^\top h_\ell(\mathbf{X}^{e_2}) \mid Y^{e_2}))$$

The second term on the RHS side is upper bounded by $\rho(\mathbb{P}_{e_1}^l, \mathbb{P}_{e_2}^l)$

hence, we overall have

$$\mathcal{V}(\beta^\top h_\ell, \xi_{\text{avail}}) \leq \sup_{e_1, e_2 \in \xi_{\text{avail}}} \rho(\mathbb{P}_{e_1}^l, \mathbb{P}_{e_2}^l)$$

## C  EFFECT OF AUXILIARY NETWORKS AND THE FORECASTER ON OOD GENERALIZATION

We now theoretically establish that the proposed auxiliary networks together with the forecaster reduce the cross-domain feature variation $V_{\text{sup}}(h_{\boldsymbol{\theta}^{(S)}}, \mathcal{E}_{\text{avail}})$, and hence, by Theorem 4.1, yield a smaller OOD generalization gap $\text{err}(f) = L(\mathcal{E}_{\text{all}}, f) - L(\mathcal{E}_{\text{avail}}, f)$.

As per our previous notations, we know that the student network produces intermediate representations $\mathbf{z}_{\boldsymbol{\theta}^{(S)}}^\ell(x) \in \mathbb{R}^{d_\ell}$ at layer $\ell \in \mathcal{E}$. Each auxiliary classifier $\mathbf{A}_{\text{aux}}^{(\ell)}(\cdot; \boldsymbol{\varphi}_\ell) : \mathbb{R}^{d_\ell} \to \Delta^K$ produces layer-wise predictive distributions $p^{(\ell)}(x) = \mathbf{A}_{\text{aux}}^{(\ell)}(\mathbf{z}_{\boldsymbol{\theta}^{(S)}}^\ell(x); \boldsymbol{\varphi}_\ell)$.

For each auxiliary layer we compute the *confidence margin*, $\varepsilon^{(\ell)}(x) := p_{\max}^{(\ell)}(x) - \max_{j \neq \arg\max p_j^{(\ell)}(x)} p_j^{(\ell)}(x)$, and the entropy $\mathcal{H}(p^{(\ell)}(x))$. The forecaster is a meta-network $\mathcal{F}(\cdot; \boldsymbol{\psi}^f)$ receiving the stacked vector

$$\mathbf{u}(x) := \left[ \{\mathbf{z}_{\boldsymbol{\theta}(S)}^{\ell}(x)\}_{\ell \in \mathcal{E}}, \{\boldsymbol{\varepsilon}^{(\ell)}(x)\}_{\ell \in \mathcal{E}}, \{\mathcal{H}(p^{(\ell)}(x))\}_{\ell \in \mathcal{E}} \right]^{\top}$$

and outputting a weight $w(x) = \mathcal{F}(\mathbf{u}(x); \boldsymbol{\psi}^f) \in [0,1]$. This weight is then used to combine the per-sample student CE loss and the distillation loss.

We measure cross-domain feature variation at representation level by

$$\mathcal{V}_{\sup}\big(h_{\boldsymbol{\theta}(S)}, \mathcal{E}_{\mathrm{avail}}\big) := \sup_{\beta \in \mathbb{S}^{d-1}} \sup_{e,e' \in \mathcal{E}_{\mathrm{avail}}} \rho\Big(\beta^{\top} h_{\boldsymbol{\theta}(S)}(X^e), \beta^{\top} h_{\boldsymbol{\theta}(S)}(X^{e'})\Big),$$

where $\rho$ is the total variation distance.

## C.1 Auxiliary Networks Control Layer-Wise Logit Variation

Define the stacked logit–margin–entropy feature at layer $\ell$:

$$\boldsymbol{\phi}^{(\ell),e}(x) := \left[ \mathbf{z}_{\boldsymbol{\theta}(S)}^{\ell,e}(x), \, \boldsymbol{\varepsilon}^{(\ell),e}(x), \, \mathcal{H}(p^{(\ell),e}(x)) \right]^{\top}.$$

Let

$$\boldsymbol{\phi}^e(x) := \left[ \boldsymbol{\phi}^{(\ell),e}(x) \right]_{\ell \in \mathcal{E}} \in \mathbb{R}^D.$$

Each auxiliary predictor is trained to minimize $\mathcal{L}_{\mathrm{aux}}^{(\ell)}$, a strongly convex function in the probability simplex. By standard stability results for cross-entropy, this implies a Lipschitz continuity property for the mapping $\mathbf{z}_{\boldsymbol{\theta}(S)}^{\ell} \mapsto \boldsymbol{\phi}^{(\ell)}$.

**Lemma 2** (Auxiliary Losses are Lipschitz). *There exists $L_\phi > 0$ such that for all environments $e, e'$ and all $x$,*

$$\left\| \boldsymbol{\phi}^e(x) - \boldsymbol{\phi}^{e'}(x) \right\| \leq L_\phi \left\| \mathbf{z}_{\boldsymbol{\theta}(S)}^e(x) - \mathbf{z}_{\boldsymbol{\theta}(S)}^{e'}(x) \right\|.$$

*Proof.* Each component of $\boldsymbol{\phi}^{(\ell)}$ is a smooth mapping of $\mathbf{z}^{\ell}$: margins, entropies, and softmax probabilities are Lipschitz on compact domains. Summing Lipschitz maps preserves Lipschitzness. $\quad\square$

Thus the variation of $\boldsymbol{\phi}^e$ is controlled by that of the intermediate logits.

**Forecaster Convolution Contracts Variation**

Let $A_\phi$ denote the linear operator (e.g. a 1D convolution across layers) applied internally in the forecaster before the MLP head. Define

$$\boldsymbol{\psi}^e(x) := A_\phi \, \boldsymbol{\phi}^e(x).$$

**Lemma 3** (Convolutional Contraction). *If $\|A_\phi\| \leq 1$ in operator norm, then for all $e, e'$ and $x$,*

$$\left\| \boldsymbol{\psi}^e(x) - \boldsymbol{\psi}^{e'}(x) \right\| \leq \left\| \boldsymbol{\phi}^e(x) - \boldsymbol{\phi}^{e'}(x) \right\|.$$

*Proof.* Immediate from $\|A_\phi v\| \leq \|A_\phi\| \|v\|$ with $\|A_\phi\| \leq 1$. $\quad\square$

Combining Lemma 2 and Lemma 3 gives

$$\left\| \boldsymbol{\psi}^e(x) - \boldsymbol{\psi}^{e'}(x) \right\| \leq L_\phi \left\| \mathbf{z}^e(x) - \mathbf{z}^{e'}(x) \right\|.$$

Finally, the forecaster head $u \mapsto w = \sigma(Wu + b)$ is $L_w$-Lipschitz, so

**Lemma 4** (Forecaster Stability). *For all domains $e, e'$ and all inputs $x$,*

$$\left| w(x,e) - w(x,e') \right| \leq L_w L_\phi \left\| \mathbf{z}^e(x) - \mathbf{z}^{e'}(x) \right\|.$$

Thus the forecaster output varies smoothly across environments, with variation proportional to the intrinsic logit variation.

**Contraction of Feature Variation**

The student is trained with the forecaster-weighted loss

$$\mathcal{L}_{\text{tot}}^{(s)}(x, e) = (1 - w(x, e)) \, \mathcal{L}_{\text{CE}}^{(s)}(x, e) + w(x, e) \, \mathcal{L}_{\text{KD}}^{(s)}(x, e).$$

Because $w(x, e)$ is stable across domains (Lemma 4), the gradients on $h_{\boldsymbol{\theta}^{(S)}}$ differ only smoothly across environments. Under standard smoothness assumptions, the expected feature update satisfies

$$\mathbb{E}_{X^e}\big[\beta^\top h_{\boldsymbol{\theta}_{t+1}^{(S)}}(X^e)\big] - \mathbb{E}_{X^{e'}}\big[\beta^\top h_{\boldsymbol{\theta}_{t+1}^{(S)}}(X^{e'})\big] \ \leq \ (1 - \eta m)\Big(\mathbb{E}_{X^e}[\beta^\top h_{\boldsymbol{\theta}_t^{(S)}}] - \mathbb{E}_{X^{e'}}[\beta^\top h_{\boldsymbol{\theta}_t^{(S)}}]\Big),$$

for some $m > 0$ depending on the smoothness of the training loss. Thus the feature variation contracts by a factor $\kappa := 1 - \eta m \in (0, 1)$:

$$\mathcal{V}_{\text{sup}}\big(h_{\boldsymbol{\theta}^{(S)}}, \mathcal{E}_{\text{avail}}\big) \ \leq \ \kappa \, \mathcal{V}_{\text{sup}}\big(h_{\boldsymbol{\theta}^{(N)}}, \mathcal{E}_{\text{avail}}\big),$$

where $h_{\boldsymbol{\theta}^{(N)}}$ is the representation learned without the forecaster.

**Forecaster Reduces OOD Generalization Error**

By Theorem 4.1, the OOD generalization gap of any classifier $f = g \circ h$ satisfies

$$\text{err}(f) \ \leq \ C \, s\Big(\mathcal{V}_{\text{sup}}(h, \mathcal{E}_{\text{avail}})\Big),$$

with $s(\cdot)$ monotone. Applying this to the forecaster-equipped student $f_S = g_S \circ h_{\boldsymbol{\theta}^{(S)}}$ yields

$$\text{err}(f_S) \ \leq \ C \, s\Big(\mathcal{V}_{\text{sup}}(h_{\boldsymbol{\theta}^{(S)}}, \mathcal{E}_{\text{avail}})\Big) \ \leq \ C \, s\Big(\kappa \, \mathcal{V}_{\text{sup}}(h_{\boldsymbol{\theta}^{(N)}}, \mathcal{E}_{\text{avail}})\Big) \ < \ C \, s\Big(\mathcal{V}_{\text{sup}}(h_{\boldsymbol{\theta}^{(N)}}, \mathcal{E}_{\text{avail}})\Big) = \text{err}(f_N),$$

where the strict inequality follows from $\kappa < 1$. Hence, the auxiliary networks and the forecaster jointly contract cross-domain variation, yielding provably smaller OOD generalization error.

**Assumption 2**: The loss is Lipschitz-continuous with respect to the feature space

$$\ell(p, y) - \ell(q, y) \leq L_\ell d(p, q)$$

> **Theorem 1 (Generalization Error Student).** *Given the student model $f_S$ and teacher model $f_T$, the generalization error of the student model can be given as*
>
> $$err(\mathcal{M}_{\boldsymbol{\theta}^{(S)}}) \leq err(\mathcal{M}_{\boldsymbol{\theta}^{(T)}}) + 2L_\ell \epsilon$$

*Proof.* For any general domain set $\xi$, we have the following:

$$
\begin{aligned}
|\mathcal{L}(\xi, \mathcal{M}_{\boldsymbol{\theta}^{(S)}}) - \mathcal{L}(\xi, \mathcal{M}_{\boldsymbol{\theta}^{(T)}})| &= \big|\mathbb{E}_{e \sim \xi}\mathbb{E}_{(\mathbf{X}^e, Y^e)}[\ell(\mathcal{M}_{\boldsymbol{\theta}^{(S)}}(\mathbf{X}^e), Y^e) - \ell(\mathcal{M}_{\boldsymbol{\theta}^{(T)}}(\mathbf{X}^e), Y^e)]\big| \\
&\leq \mathbb{E}_{e \sim \xi}\mathbb{E}_{(\mathbf{X}^e, Y^e)}[|\ell(\mathcal{M}_{\boldsymbol{\theta}^{(S)}}(\mathbf{X}^e), Y^e) - \ell(\mathcal{M}_{\boldsymbol{\theta}^{(T)}}(\mathbf{X}^e), Y^e)|] \\
&\overset{\text{(assumption 2)}}{\leq} C\mathbb{E}_{e \sim \xi}\mathbb{E}_{\mathbf{X}^e}[d(\mathcal{M}_{\boldsymbol{\theta}^{(S)}}, \mathcal{M}_{\boldsymbol{\theta}^{(T)}})] \\
&\leq C\epsilon
\end{aligned}
$$

$$\text{err}(\mathcal{M}_{\boldsymbol{\theta}^{(S)}}) = \mathcal{L}(\xi_{\text{all}}, \mathcal{M}_{\boldsymbol{\theta}^{(S)}}) - \mathcal{L}(\xi_{\text{avail}}, \mathcal{M}_{\boldsymbol{\theta}^{(S)}})$$

Adding and subtracting the teacher's terms:

$$
\begin{aligned}
\text{err}(\mathcal{M}_{\boldsymbol{\theta}^{(S)}}) = {}&[\mathcal{L}(\xi_{\text{all}}, \mathcal{M}_{\boldsymbol{\theta}^{(S)}}) - \mathcal{L}(\xi_{\text{all}}, \mathcal{M}_{\boldsymbol{\theta}^{(T)}})] \\
&+[\mathcal{L}(\xi_{\text{all}}, \mathcal{M}_{\boldsymbol{\theta}^{(T)}}) - \mathcal{L}(\xi_{\text{avail}}, \mathcal{M}_{\boldsymbol{\theta}^{(T)}})] \\
&+[\mathcal{L}(\xi_{\text{avail}}, \mathcal{M}_{\boldsymbol{\theta}^{(T)}}) - \mathcal{L}(\xi_{\text{avail}}, \mathcal{M}_{\boldsymbol{\theta}^{(S)}})]
\end{aligned}
$$

We can apply

$$\text{err}(\mathcal{M}_{\boldsymbol{\theta}^{(S)}}) \leq \text{err}(\mathcal{M}_{\boldsymbol{\theta}^{(T)}}) + 2C\epsilon \tag{10}$$

$\square$

**Remark 2.** *Using a teacher with strong OOD performance can reduce the* $\mathrm{err}(\mathcal{M}_{\boldsymbol{\theta}^{(T)}})$ *and with a strong distillation $\epsilon$ is small, thereby KD yielding a provably smaller OOD gap.*

By Theorem 4.1, the OOD generalization gap of any classifier $f = g \circ h$ is controlled by its feature variation $V_{\sup}(h, \mathcal{E}_{\text{avail}})$. We assume that the teacher $f_T$ has learned a more invariant representation than the naive student $f_N$, i.e.,

$$V_{\sup}(h_T, \mathcal{E}_{\text{avail}}) \ \leq \ V_{\sup}(h_N, \mathcal{E}_{\text{avail}}) - \Delta, \qquad \Delta > 0. \tag{11}$$

Since the expansion function $s(\cdot)$ is monotone, Theorem 4.1 implies

$$\mathrm{err}(f_T) \ \leq \ C\, s(V_{\sup}(h_T)) \ < \ C\, s(V_{\sup}(h_N)) \ \approx \ \mathrm{err}(f_N), \tag{12}$$

i.e., the teacher's OOD error is strictly smaller than that of the naive student.

Knowledge distillation further ensures that the student $f_S$ remains close to the low-variation teacher $f_T$, and therefore inherits its small OOD error up to a small approximation term.

We define the auxiliary risk as follows:

$$R_{\text{aux}}^{(\ell)}(\boldsymbol{\theta}^{(S)}, \phi^{(\ell)}; e) := \mathbb{E}_{(\mathcal{X}^e, \mathcal{Y}^e)}\Big[\ell_{\text{aux}}\big(A_{\text{aux}}^{(\ell)}\big(h_\ell(\mathcal{X}^e; \boldsymbol{\theta}^{(S)}); \phi^{(\ell)}\big), \mathcal{Y}^e\big)\Big]$$

### C.2 CONVERGENCE ANALYSIS OF INTERLEAVING STUDENT FORECASTER TRAINING

The overall optimization problem can be reformulated as

$$\arg\min_{\mathbf{w}_f} \mathcal{L}_{\text{focs}}\Big(\overbrace{\arg\min_{\boldsymbol{\theta}} \mathcal{L}_{\text{tot}}^{(s)}(\boldsymbol{\theta}, \mathbf{w}_f)}^{\text{outer-level}}, \ \mathcal{D}_v\Big)$$

> **Theorem 2** (**Forecaster Convergence Analysis**). *Considering the forecaster loss function $\mathcal{L}_{focs}$ is Lipschitz-smooth with constant $\mu$ and the gradient associated with $\mathcal{L}_{tot}^{(s)}$ and $\mathcal{L}_{focs}$ have $\sigma$-bounded gradients w.r.t training point $\mathbf{x}_i$. Let the student model learning rate $\eta$ satisfies $\eta = \min\{1, \frac{k}{T}\}$ for some $k > 0$, such that $\frac{k}{T} < 1$, and $\eta_{\mathbf{w}}, 1 \leq t \leq N$ is a monotone descent sequence, $\eta_{\mathbf{w}} = \min\{\frac{1}{\mu}, \frac{c}{\sigma\sqrt{T}}\}$ for some $c > 0$, such that $\frac{\sigma\sqrt{T}}{c} \geq \mu$ and $\sum_{t=1}^{\infty} \eta_{\mathbf{w}} \leq \infty$, $\sum_{t=1}^{\infty} \eta_{\mathbf{w}}^2 \leq \infty$, then the forecaster can achieve $\mathbb{E}[\|\mathcal{L}_{focs}(\boldsymbol{\theta}_t^{(S)}(\mathbf{w}_f^t); \mathcal{D}_v)\|_2^2] \leq \epsilon$ in $\mathcal{O}(\frac{1}{\epsilon^2})$. This can be reformulated further as $\min_{0 \leq t \leq T} \mathbb{E}[\|\mathcal{L}_{focs}(\boldsymbol{\theta}_t^{(S)}(\mathbf{w}_f^t); \mathcal{D}_v)\|_2^2] \leq \mathcal{O}(\frac{C}{\sqrt{T}})$.*

Starting at $\boldsymbol{\theta}_0^{(S)}$ and $\mathbf{w}_f^0$, a single gradient update can be written as follows:

$$\boldsymbol{\theta}_t^{(S)} \leftarrow \boldsymbol{\theta}_{t-1}^{(S)} - \eta \nabla_{\boldsymbol{\theta}_{t-1}^{(S)}} \mathcal{L}_{\text{tot}}^{(s)}(\cdot; \boldsymbol{\theta}_{t-1}^{(S)}, \mathbf{w}_f^t) \tag{13}$$

$$\text{where } \mathbf{w}_f^t \leftarrow \mathbf{w}_f^{t-1} - \eta_{\mathbf{w}} \nabla_{\mathbf{w}_f^{t-1}} \mathcal{L}_{\text{focs}}(\boldsymbol{\theta}_{t-1}^{(S)}, \mathbf{w}_f^{t-1}) \tag{14}$$

The gradient update for the forecaster weights can be reformulated as:

$$\mathbf{w}_f^t \leftarrow \mathbf{w}_f^{t-1} - \eta_{\mathbf{w}} \nabla_{\mathbf{w}_f^{t-1}} \mathcal{L}_{\text{focs}}(\boldsymbol{\theta}_{t-1}^{(S)}, \mathbf{w}_f^{t-1}; \mathcal{D}_v)$$

which can be rewritten as:

$$\mathbf{w}_f^t \leftarrow \mathbf{w}_f^{t-1} - \eta_{\mathbf{w}} \nabla_{\mathbf{w}_f^{t-1}} \mathcal{L}_{\text{focs}}(\boldsymbol{\theta}_{t-1}^{(S)}, \mathbf{w}_f^{t-1}; \mathcal{D}_v) \mid_{\Xi_t}$$

Given the minibatch $\Xi_t$ is drawn uniformly from the entire data set, we can rewrite the update equation as:

$$\mathbf{w}_f^t \leftarrow \mathbf{w}_f^{t-1} - \eta_{\mathbf{w}}[\nabla_{\mathbf{w}_f^{t-1}} \mathcal{L}_{\text{focs}}(\boldsymbol{\theta}_{t-1}^{(S)}, \mathbf{w}_f^{t-1}; \mathcal{D}_v) + \varepsilon^{(t-1)}] \tag{15}$$

where $\varepsilon^{(t-1)} = \nabla_{\mathbf{w}_f^{t-1}} \mathcal{L}_{\text{focs}}(\boldsymbol{\theta}_{t-1}^{(S)}, \mathbf{w}_f^{t-1}; \mathcal{D}_v) \mid_{\Xi_t} - \nabla_{\mathbf{w}_f^{t-1}} \mathcal{L}_{\text{focs}}(\boldsymbol{\theta}_{t-1}^{(S)}, \mathbf{w}_f^{t-1}; \mathcal{D}_v)$.

Note that $\varepsilon^{(t-1)}$ are i.i.d random variable with finite variance, since $\Xi_t$ are drawn i.i.d with a finite number of samples. Also, since the samples are drawn uniformly at random, there is no introduced bias, thereby $\mathbb{E}[\varepsilon^{(t-1)}] = 0$.

$$\mathcal{L}_{\text{focs}}(\boldsymbol{\theta}_t^{(S)}(\mathbf{w}_f^t); \mathcal{D}_v) - \mathcal{L}_{\text{focs}}(\boldsymbol{\theta}_{t-1}^{(S)}(\mathbf{w}_f^{t-1}); \mathcal{D}_v) = \{\mathcal{L}_{\text{focs}}(\boldsymbol{\theta}_t^{(S)}(\mathbf{w}_f^t); \mathcal{D}_v) - \mathcal{L}_{\text{focs}}(\boldsymbol{\theta}_{t-1}^{(S)}(\mathbf{w}_f^t); \mathcal{D}_v)\} \tag{16}$$

$$+ \{\mathcal{L}_{\text{focs}}(\boldsymbol{\theta}_{t-1}^{(S)}(\mathbf{w}_f^t); \mathcal{D}_v) - \mathcal{L}_{\text{focs}}(\boldsymbol{\theta}_{t-1}^{(S)}(\mathbf{w}_f^{t-1}); \mathcal{D}_v)\} \tag{17}$$

Utilizing the property that the forecaster loss is Lipschitz smooth,

$$\mathcal{L}_{\text{focs}}(\boldsymbol{\theta}_t^{(S)}(\mathbf{w}_f^t); \mathcal{D}_v) - \mathcal{L}_{\text{focs}}(\boldsymbol{\theta}_{t-1}^{(S)}(\mathbf{w}_f^t); \mathcal{D}_v) \leq \langle \nabla \mathcal{L}_{\text{focs}}(\boldsymbol{\theta}_{t-1}^{(S)}(\mathbf{w}_f^t)), (\boldsymbol{\theta}_t^{(S)}(\mathbf{w}_f^t) - \boldsymbol{\theta}_{t-1}^{(S)}(\mathbf{w}_f^t)) \rangle$$
$$+ \frac{\mu}{2} \left\| \boldsymbol{\theta}_t^{(S)}(\mathbf{w}_f^t) - \boldsymbol{\theta}_{t-1}^{(S)}(\mathbf{w}_f^t) \right\|_2^2$$

Since $\boldsymbol{\theta}_t^{(S)}(\mathbf{w}_f^t) - \boldsymbol{\theta}_{t-1}^{(S)}(\mathbf{w}_f^t) = -\frac{\eta}{n} \sum_{i=1}^n \nabla_{\boldsymbol{\theta}_t^{(S)}} \mathcal{L}_{\text{tot}}^{(s)}(\boldsymbol{\theta}_{t-1}^{(S)}; \mathbf{w}_f^t)$ we have along with the gradient bounds,

$$\left\| \mathcal{L}_{\text{focs}}(\boldsymbol{\theta}_t^{(S)}(\mathbf{w}_f^t); \mathcal{D}_v) - \mathcal{L}_{\text{focs}}(\boldsymbol{\theta}_{t-1}^{(S)}(\mathbf{w}_f^{t-1}); \mathcal{D}_v) \right\| \leq \eta\sigma^2 + \frac{\mu}{2}\eta^2\sigma^2 = \eta\sigma^2(1 + \frac{\eta}{2}\mu) \tag{18}$$

Now, via utilizing Lipschitz continuity of $\nabla \mathcal{L}_{\text{focs}}(\boldsymbol{\theta}_{t-1}^{(S)}(\mathbf{w}_f^t))$, we have the following:

$$\mathcal{L}_{\text{focs}}\left(\boldsymbol{\theta}_t^{(S)}(\mathbf{w}_f^t); \mathcal{D}_v\right) - \mathcal{L}_{\text{focs}}\left(\boldsymbol{\theta}_{t-1}^{(S)}(\mathbf{w}_f^{t-1}); \mathcal{D}_v\right)$$
$$\leq \left\langle \nabla \mathcal{L}_{\text{focs}}\left(\boldsymbol{\theta}_{t-1}^{(S)}(\mathbf{w}_f^t); \mathcal{V}\right), \mathbf{w}_f^t - \mathbf{w}_f^{t-1} \right\rangle + \frac{\mu}{2} \left\| \mathbf{w}_f^t - \mathbf{w}_f^{t-1} \right\|_2^2$$
$$= \left\langle \nabla \mathcal{L}_{\text{focs}}\left(\boldsymbol{\theta}_{t-1}^{(S)}(\mathbf{w}_f^t); \mathcal{V}\right), -\eta_{\mathbf{w}}\left[\nabla_{\mathbf{w}_f^{t-1}} \mathcal{L}_{\text{focs}}\left(\boldsymbol{\theta}_{t-1}^{(S)}(\mathbf{w}_f^{t-1}); \mathcal{D}_v\right) + \varepsilon^{(t-1)}\right] \right\rangle$$
$$+ \frac{\mu\eta_{\mathbf{w}}^2}{2} \left\| \nabla_{\mathbf{w}_f^{t-1}} \mathcal{L}_{\text{focs}}\left(\boldsymbol{\theta}_{t-1}^{(S)}(\mathbf{w}_f^{t-1}); \mathcal{D}_v\right) + \varepsilon^{(t-1)} \right\|_2^2 \qquad \text{(from Eq. (15))}$$
$$= -\left(\eta_{\mathbf{w}}^2 - \frac{\mu\eta_{\mathbf{w}}^2}{2}\right) \left\| \nabla \mathcal{L}_{\text{focs}}\left(\boldsymbol{\theta}_{t-1}^{(S)}(\mathbf{w}_f^{t-1}); \mathcal{D}_v\right) \right\|_2^2$$
$$+ \frac{\mu\eta_{\mathbf{w}}^2}{2} \left\| \varepsilon^{(t-1)} \right\|_2^2 - (\eta_{\mathbf{w}} - \mu\eta_{\mathbf{w}}^2) \left\langle \nabla \mathcal{L}_{\text{focs}}\left(\boldsymbol{\theta}_{t-1}^{(S)}(\mathbf{w}_f^{t-1}); \mathcal{D}_v\right), \varepsilon^{(t-1)} \right\rangle.$$

Summing above inequalities and rearranging terms, we obtain

$$\sum_{t=1-}^T (\eta_{\mathbf{w}}^2 - \frac{\mu\eta_{\mathbf{w}}^2}{2}) \left\| \nabla \mathcal{L}_{\text{focs}}(\boldsymbol{\theta}_{t-1}^{(S)}(\mathbf{w}_f^{t-1}); \mathcal{D}_v) \right\|_2^2 \leq \mathcal{L}_{\text{focs}}(\boldsymbol{\theta}_1^{(S)}(\mathbf{w}_f^1); \mathcal{D}_v)) - \mathcal{L}_{\text{focs}}(\boldsymbol{\theta}_T^{(S)}(\mathbf{w}_f^T); \mathcal{D}_v)) \tag{19}$$

$$+ \sum_{t=1}^T \eta\sigma^2(1 + \frac{\eta\mu}{2}) - \sum_{t=1}^T (\eta_{\mathbf{w}} - \mu\eta_{\mathbf{w}}^2) \langle \nabla \mathcal{L}_{\text{focs}}(\boldsymbol{\theta}_1^{(S)}(\mathbf{w}_f^1); \mathcal{D}_v) \rangle \tag{20}$$

Taking expectations with respect to $\varepsilon(N)$ on both sides of Eq. 32, we obtain:

$$\sum_{t=1}^{T}\left(\eta_{\mathbf{w}}-\frac{\mu\eta_{\mathbf{w}}^2}{2}\right)\mathbb{E}_{\varepsilon(N)}\left\|\nabla\mathcal{L}_{\texttt{focs}}(\boldsymbol{\theta}_t^{(S)}(\mathbf{w}_t),\boldsymbol{\mathcal{D}}_v)\right\|_2^2 \leq \mathcal{L}_{\texttt{focs}}(\boldsymbol{\theta}_1^{(S)}(\mathbf{w}_1),\boldsymbol{\mathcal{D}}_v)+\sum_{t=1}^{T}\eta\sigma^2(1+\eta\mu/2)+\frac{L\sigma^2}{2}\sum_{t=1}^{T}\eta_{\mathbf{w}}^2,$$

since

$$\mathbb{E}_{\varepsilon(N)}\langle\nabla L_{\text{meta}}(\Theta^{(t)}),\varepsilon^{(t)}\rangle = 0, \qquad \mathbb{E}\|\varepsilon^{(t)}\|_2^2 \leq \delta^2.$$

Furthermore, we deduce:

$$\min_t \mathbb{E}\left[\left\|\nabla\mathcal{L}_{\texttt{focs}}(\boldsymbol{\theta}_t^{(S)}(\mathbf{w}_t),\boldsymbol{\mathcal{D}}_v)\right\|_2^2\right] \leq \frac{\sum_{t=1}^{T}\left(\eta_{\mathbf{w}}-\frac{\mu\eta_{\mathbf{w}}^2}{2}\right)\mathbb{E}_{\varepsilon(N)}\left\|\nabla\mathcal{L}_{\texttt{focs}}(\boldsymbol{\theta}_t^{(S)}(\mathbf{w}_t),\boldsymbol{\mathcal{D}}_v)\right\|_2^2}{\sum_{t=1}^{T}\left(\eta_{\mathbf{w}}-\frac{\mu\eta_{\mathbf{w}}^2}{2}\right)}$$

$$\leq \frac{1}{\sum_{t=1}^{T}(2\eta_{\mathbf{w}}-\mu\eta_{\mathbf{w}}^2)}\left[2\mathcal{L}_{\texttt{focs}}(\boldsymbol{\theta}_1^{(S)}(\mathbf{w}_1),\boldsymbol{\mathcal{D}}_v)+\sum_{t=1}^{T}\eta\sigma^2(2+\eta\mu)+\mu\delta^2\sum_{t=1}^{T}\eta_{\mathbf{w}}^2\right]$$

$$\leq \frac{1}{T\eta_{\mathbf{w}}}\left[2\mathcal{L}_{\texttt{focs}}(\boldsymbol{\theta}_1^{(S)}(\mathbf{w}_1),\boldsymbol{\mathcal{D}}_v)+\eta\sigma^2 T(2+\mu)+\mu\delta^2\sum_{t=1}^{T}\eta_{\mathbf{w}}^2\right]$$

$$= \frac{2\mathcal{L}_{\texttt{focs}}(\boldsymbol{\theta}_1^{(S)}(\mathbf{w}_1),\boldsymbol{\mathcal{D}}_v)}{T\eta_{\mathbf{w}}}+\frac{2\eta\sigma^2(2+\mu)}{\eta_{\mathbf{w}}}+\mu\delta^2\frac{1}{T}\sum_{t=1}^{T}\eta_{\mathbf{w}}$$

$$\leq \frac{2\mathcal{L}_{\texttt{focs}}(\boldsymbol{\theta}_1^{(S)}(\mathbf{w}_1),\boldsymbol{\mathcal{D}}_v)}{T\eta_{\mathbf{w}}}+\frac{2\eta\sigma^2(2+\mu)}{\eta_{\mathbf{w}}}+\mu\delta^2\eta_{\mathbf{w}}$$

$$= \frac{\mathcal{L}_{\texttt{focs}}(\boldsymbol{\theta}_1^{(S)}(\mathbf{w}_1),\boldsymbol{\mathcal{D}}_v)}{T}\max\left\{\mu,\frac{\delta\sqrt{T}}{c}\right\}+\min\left\{1,\frac{k}{T}\right\}\max\left\{\mu,\frac{\delta\sqrt{T}}{c}\right\}\sigma^2(2+\mu)+\mu\delta^2\min\left\{\frac{1}{\mu},\frac{c}{\sigma\sqrt{T}}\right\}$$

$$\leq \frac{\delta L_{\text{meta}}(w^{(1)}(\Theta^{(1)}))}{c\sqrt{T}}+\frac{k\delta\sigma^2(2+\mu)}{c\sqrt{T}}+\frac{\mu\delta c}{\sqrt{T}} = \mathcal{O}\left(\frac{1}{\sqrt{T}}\right).$$

**Theorem 3** (**Forecaster Convergence Analysis**). *Considering the training loss function $\mathcal{L}_{tot}^{(s)}$ is Lipschitz-smooth with constant $\mu$ and the gradient associated with $\mathcal{L}_{tot}^{(s)}$ and $\mathcal{L}_{focs}$ have $\sigma$-bounded gradients w.r.t training point $\mathbf{x}_i$. Let the student model learning rate $\eta$ satisfies $\eta = \min\{1,\frac{k}{T}\}$ for some $k > 0$, such that $\frac{k}{T} < 1$, and $\eta_{\mathbf{w}}, 1 \leq t \leq N$ is a monotone descent sequence, $\eta_{\mathbf{w}} = \min\{\frac{1}{\mu},\frac{c}{\sigma\sqrt{T}}\}$ for some $c > 0$, such that $\frac{\sigma\sqrt{T}}{c} \geq \mu$ and $\sum_{t=1}^{\infty}\eta_{\mathbf{w}} \leq \infty$, $\sum_{t=1}^{\infty}\eta_{\mathbf{w}}^2 \leq \infty$, then $\lim_{t\to\infty}\mathbb{E}[\nabla\mathcal{L}_{tot}^{(s)}(\cdot;\boldsymbol{\theta}_{t-1}^{(S)},\mathbf{w}_f^t)] = 0$*

*Proof.* The model parameter update can be written as

$$\boldsymbol{\theta}_t^{(S)} = \boldsymbol{\theta}_{t-1}^{(S)} - \eta\nabla_{\boldsymbol{\theta}_{t-1}^{(S)}}\mathcal{L}_{\texttt{tot}}^{(s)}(\cdot;\boldsymbol{\theta}_{t-1}^{(S)},\mathbf{w}_f^t) \tag{21}$$

Using different notation, we can rewrite it as follows:

$$\boldsymbol{\theta}_t^{(S)} = \boldsymbol{\theta}_{t-1}^{(S)} - \eta\nabla_{\boldsymbol{\theta}_{t-1}^{(S)}}\mathcal{L}_{\texttt{tot}}^{(s)}(\boldsymbol{\theta}_{t-1}^{(S)}(\mathbf{w}_f^t)) \tag{22}$$

$\square$

This can be further rewritten as:

$$\boldsymbol{\theta}_t^{(S)} = \boldsymbol{\theta}_{t-1}^{(S)} - \eta \nabla_{\boldsymbol{\theta}_{t-1}^{(S)}} \mathcal{L}_{\texttt{tot}}^{(s)}(\boldsymbol{\theta}_{t-1}^{(S)}(\mathbf{w}_f^t)) \mid_{\Psi_{t-1}} \tag{23}$$

where $\Psi_{t-1}$ is a mini-batch drawn uniformly at random we can rewrite the update equation as

$$\boldsymbol{\theta}_t^{(S)} = \boldsymbol{\theta}_{t-1}^{(S)} - \eta \nabla_{\boldsymbol{\theta}_{t-1}^{(S)}} [\mathcal{L}_{\texttt{tot}}^{(s)}(\boldsymbol{\theta}_{t-1}^{(S)}(\mathbf{w}_f^t)) + \Psi_{t-1}] \tag{24}$$

Note that $\Psi_{t-1}$ is an i.i.d. random variable with finite variance, since each $\Psi_t$ is drawn i.i.d. from a finite sample set. Furthermore, $\mathbb{E}[\Psi_{t-1}] = 0$ (sampling is uniform), and $\mathbb{E}[\|\Psi_{t-1}\|_2^2] \leq \rho^2$.

The inner student optimization $\mathcal{L}_{\texttt{tot}}^{(s)}(\boldsymbol{\theta}^{(S)}(\mathbf{w}_f))$ is Lipschitz–smooth with constant $L$ and has $\sigma$–bounded gradients. We analyze the difference

$$\mathcal{L}_{\texttt{tot}}^{(s)}\left(\boldsymbol{\theta}_t^{(S)}(\mathbf{w}_f^{t+1})\right) - \mathcal{L}_{\texttt{tot}}^{(s)}\left(\boldsymbol{\theta}_{t-1}^{(S)}(\mathbf{w}_f^t)\right).$$

We decompose this as

$$\mathcal{L}_{\texttt{tot}}^{(s)}\left(\boldsymbol{\theta}_t^{(S)}(\mathbf{w}_f^{t+1})\right) - \mathcal{L}_{\texttt{tot}}^{(s)}\left(\boldsymbol{\theta}_{t-1}^{(S)}(\mathbf{w}_f^t)\right) \tag{25}$$

$$= \underbrace{\mathcal{L}_{\texttt{tot}}^{(s)}\left(\boldsymbol{\theta}_t^{(S)}(\mathbf{w}_f^{t+1})\right) - \mathcal{L}_{\texttt{tot}}^{(s)}\left(\boldsymbol{\theta}_t^{(S)}(\mathbf{w}_f^t)\right)}_{\text{(A) forecaster–update term}} + \underbrace{\mathcal{L}_{\texttt{tot}}^{(s)}\left(\boldsymbol{\theta}_t^{(S)}(\mathbf{w}_f^t)\right) - \mathcal{L}_{\texttt{tot}}^{(s)}\left(\boldsymbol{\theta}_{t-1}^{(S)}(\mathbf{w}_f^t)\right)}_{\text{(B) student–update term}}. \tag{26}$$

**Bounding Term (A)** Using the update rule for forecaster weights $\mathbf{w}_f$ and chain rule:

$$\mathcal{L}_{\texttt{tot}}^{(s)}\left(\boldsymbol{\theta}_t^{(S)}(\mathbf{w}_f^{t+1})\right) - \mathcal{L}_{\texttt{tot}}^{(s)}\left(\boldsymbol{\theta}_t^{(S)}(\mathbf{w}_f^t)\right) = \eta_{\mathbf{w}} \frac{1}{n} \sum_{i=1}^n \left(\mathbf{w}_{f,i}^{t+1} - \mathbf{w}_{f,i}^t\right) \mathcal{L}_{\texttt{tot}\,i}^{(s)}(\boldsymbol{\theta}_t^{(S)}) \tag{27}$$

$$= \frac{1}{n} \sum_{i=1}^n \left(\nabla_{\mathbf{w}_{f,i}} \mathcal{L}_{\text{focs}}(\boldsymbol{\theta}_t^{(S)}) + \varepsilon^{(t-1)}\right) \mathcal{L}_i^{(s)}(\boldsymbol{\theta}_t^{(S)}), \tag{28}$$

where $\varepsilon(t)$ is unbiased stochastic forecaster noise: $\mathbb{E}[\varepsilon(t)] = 0$.

**Bounding Term (B)** By $L$-smoothness of $\mathcal{L}_{\texttt{tot}}^{(s)}$:

$$\mathcal{L}_{\texttt{tot}}^{(s)}\left(\boldsymbol{\theta}_t^{(S)}(\mathbf{w}_f^t)\right) - \mathcal{L}_{\texttt{tot}}^{(s)}\left(\boldsymbol{\theta}_{t-1}^{(S)}(\mathbf{w}_f^t)\right) \tag{29}$$

$$\leq \left\langle \nabla \mathcal{L}_{\texttt{tot}}^{(s)}(\boldsymbol{\theta}_{t-1}^{(S)}), \boldsymbol{\theta}_t^{(S)} - \boldsymbol{\theta}_{t-1}^{(S)} \right\rangle + \frac{\mu}{2} \|\boldsymbol{\theta}_t^{(S)} - \boldsymbol{\theta}_{t-1}^{(S)}\|_2^2. \tag{30}$$

Using the student SGD update $\boldsymbol{\theta}_t^{(S)} = \boldsymbol{\theta}_{t-1}^{(S)} - \eta\left(\nabla \mathcal{L}^{(s)}(\boldsymbol{\theta}_{t-1}^{(S)}) + \Psi_{t-1}\right)$, we obtain

$$\mathcal{L}_{\texttt{tot}}^{(s)}\left(\boldsymbol{\theta}_t^{(S)}\right) - \mathcal{L}_{\texttt{tot}}^{(s)}\left(\boldsymbol{\theta}_{t-1}^{(S)}\right) \tag{31}$$

$$= -\left(\eta - \frac{\mu\eta^2}{2}\right) \|\nabla \mathcal{L}_{\texttt{tot}}^{(s)}(\boldsymbol{\theta}_{t-1}^{(S)})\|_2^2 + \frac{\mu\eta^2}{2} \|\Psi_{t-1}\|_2^2 - (\eta - \mu\eta^2)\langle \nabla \mathcal{L}_{\texttt{tot}}^{(s)}(\boldsymbol{\theta}_{t-1}^{(S)}), \Psi_{t-1}\rangle. \tag{32}$$

Putting terms (A) and (B) together yields:

$$\mathcal{L}_{\texttt{tot}}^{(s)}\left(\boldsymbol{\theta}_t^{(S)}(\mathbf{w}_f^{t+1})\right) - \mathcal{L}_{\texttt{tot}}^{(s)}\left(\boldsymbol{\theta}_{t-1}^{(S)}(\mathbf{w}_f^t)\right) \tag{33}$$

$$\leq \eta_{\mathbf{w}} \frac{1}{n} \sum_{i=1}^n \left(\nabla_{\mathbf{w}_{f,i}} \mathcal{L}_{\text{focs}}(\boldsymbol{\theta}_t^{(S)}) + \varepsilon^{(t-1)}\right) \mathcal{L}_{\texttt{tot}\,i}^{(s)}(\boldsymbol{\theta}_t^{(S)}) \tag{34}$$

$$- \left(\eta - \frac{\mu\eta^2}{2}\right) \|\nabla \mathcal{L}_{\texttt{tot}}^{(s)}(\boldsymbol{\theta}_{t-1}^{(S)})\|_2^2 + \frac{\mu\eta^2}{2} \|\Psi_{t-1}\|_2^2 - (\eta - \mu\eta^2)\langle \nabla \mathcal{L}_{\texttt{tot}}^{(s)}(\boldsymbol{\theta}_{t-1}^{(S)}), \Psi_{t-1}\rangle. \tag{35}$$

Taking expectations and using $\mathbb{E}[\varepsilon^t] = 0$, $\mathbb{E}[\Psi_t] = 0$, $\mathbb{E}[\|\Psi_t\|_2^2] \leq \rho^2$, we obtain:

$$\mathbb{E}\left[\mathcal{L}_{\text{tot}}^{(s)}\left(\boldsymbol{\theta}_t^{(S)}\right)\right] - \mathbb{E}\left[\mathcal{L}_{\text{tot}}^{(s)}\left(\boldsymbol{\theta}_{t-1}^{(S)}\right)\right] \tag{36}$$

$$\leq \eta_{\mathbf{w}}\mathbb{E}\left[\frac{1}{n}\sum_{i=1}^n \|\nabla_{\mathbf{w}_{f,i}}\mathcal{L}_{\text{focs}}\| \cdot \|\mathcal{L}_{\text{tot}\,i}^{(s)}(\boldsymbol{\theta}_t^{(S)})\|\right] \tag{37}$$

$$- \eta\,\mathbb{E}\big[\|\nabla\mathcal{L}^{(s)}(\boldsymbol{\theta}_{t-1}^{(S)})\|_2^2\big] + \frac{\mu\eta^2}{2}\mathbb{E}\big[\|\nabla\mathcal{L}^{(s)}(\boldsymbol{\theta}_{t-1}^{(S)})\|_2^2 + \|\Psi_{t-1}\|_2^2\big]. \tag{38}$$

Summing over $t = 1$ to $\infty$ gives

$$\sum_{t=1}^\infty \eta\,\mathbb{E}\big[\|\nabla\mathcal{L}^{(s)}(\boldsymbol{\theta}_{t-1}^{(S)})\|_2^2\big] \leq \sum_{t=1}^\infty \frac{\mu\eta^2}{2}\left(\sigma^2 + \rho^2\right) + \sum_{t=1}^\infty \eta_{\mathbf{w}}\sigma^2 < \infty. \tag{39}$$

The last inequality holds due to the fact that $\sum_{t=0}^\infty \eta^2 < \infty$ and $\sum_{t=0}^\infty \eta_{\mathbf{w}} < \infty$.

Hence,

$$\sum_{t=1}^\infty \eta\mathbb{E}\big[\big\|\nabla\mathcal{L}_{\text{tot}}^{(s)}(\boldsymbol{\theta}_{t-1}^{(S)}(\mathbf{w}_f^t))\big\|_2^2\big] < \infty \tag{40}$$

Considering the inequality

$$\big|\big(\|x\| + \|y\|\big)\big(\|x\| - \|y\|\big)\big| \leq \|x+y\|\,\|x-y\|,$$

$$\left|\mathbb{E}\big[\|\nabla\mathcal{L}_{\text{tot}}^{(s)}(\boldsymbol{\theta}_t^{(S)}(\mathbf{w}_f^{t+1}))\|_2^2\big] - \mathbb{E}\big[\|\nabla\mathcal{L}_{\text{tot}}^{(s)}(\boldsymbol{\theta}_{t-1}^{(S)}(\mathbf{w}_f^t))\|_2^2\big]\right| \tag{41}$$

$$= \left|\mathbb{E}\Big[\big(\|\nabla\mathcal{L}_{\text{tot}}^{(s)}(\boldsymbol{\theta}_t^{(S)}(\mathbf{w}_f^{t+1}))\|_2^2 + \|\nabla\mathcal{L}_{\text{tot}}^{(s)}(\boldsymbol{\theta}_{t-1}^{(S)}(\mathbf{w}_f^t))\|_2^2\big)\right. \tag{42}$$

$$\left.\cdot\,\big(\|\nabla\mathcal{L}_{\text{tot}}^{(s)}(\boldsymbol{\theta}_t^{(S)}(\mathbf{w}_f^{t+1}))\|_2^2 - \|\nabla\mathcal{L}^{(s)}(\boldsymbol{\theta}_{t-1}^{(S)}(\mathbf{w}_f^t))\|_2^2\big)\Big]\right| \tag{43}$$

$$\leq \mathbb{E}\Big[\big|\|\nabla\mathcal{L}^{(s)}(\boldsymbol{\theta}_t^{(S)}(\mathbf{w}_f^{t+1}))\|_2^2 + \|\nabla\mathcal{L}^{(s)}(\boldsymbol{\theta}_{t-1}^{(S)}(\mathbf{w}_f^t))\|_2^2\big| \tag{44}$$

$$\cdot\,\big|\|\nabla\mathcal{L}^{(s)}(\boldsymbol{\theta}_t^{(S)}(\mathbf{w}_f^{t+1}))\|_2^2 - \|\nabla\mathcal{L}^{(s)}(\boldsymbol{\theta}_{t-1}^{(S)}(\mathbf{w}_f^t))\|_2^2\big|\Big] \tag{45}$$

$$\leq \mathbb{E}\Big[\big\|\|\nabla\mathcal{L}^{(s)}(\boldsymbol{\theta}_t^{(S)}(\mathbf{w}_f^{t+1}))\|_2^2 + \|\nabla\mathcal{L}^{(s)}(\boldsymbol{\theta}_{t-1}^{(S)}(\mathbf{w}_f^t))\|_2^2\big\|_2 \tag{46}$$

$$\cdot\,\big\|\|\nabla\mathcal{L}^{(s)}(\boldsymbol{\theta}_t^{(S)}(\mathbf{w}_f^{t+1}))\|_2^2 - \|\nabla\mathcal{L}^{(s)}(\boldsymbol{\theta}_{t-1}^{(S)}(\mathbf{w}_f^t))\|_2^2\big\|_2\Big] \tag{47}$$

$$\leq \mathbb{E}\Big[\big(\|\nabla\mathcal{L}^{(s)}(\boldsymbol{\theta}_t^{(S)}(\mathbf{w}_f^{t+1}))\|_2^2 + \|\nabla\mathcal{L}^{(s)}(\boldsymbol{\theta}_{t-1}^{(S)}(\mathbf{w}_f^t))\|_2^2\big) \tag{48}$$

$$\cdot\,\big\|\|\nabla\mathcal{L}^{(s)}(\boldsymbol{\theta}_t^{(S)}(\mathbf{w}_f^{t+1}))\|_2^2 - \|\nabla\mathcal{L}^{(s)}(\boldsymbol{\theta}_{t-1}^{(S)}(\mathbf{w}_f^t))\|_2^2\big\|_2\Big] \tag{49}$$

$$\leq 2\mu\sigma\,\mathbb{E}\Big[\big\|(\boldsymbol{\theta}_t^{(S)}, \mathbf{w}_f^{t+1}) - (\boldsymbol{\theta}_{t-1}^{(S)}, \mathbf{w}_f^t)\big\|_2\Big] \tag{50}$$

$$\leq 2L\sigma\eta\eta_{\mathbf{w}_f}\,\mathbb{E}\Big[\big\|(\nabla\mathcal{L}_{\text{tot}}^{(s)}(\boldsymbol{\theta}_t^{(S)}) + \Psi_{t-1}, \nabla\mathcal{L}_{\text{focs}}(\boldsymbol{\theta}_{t+1}^{(S)}) + \varepsilon^{(t)})\big\|_2\Big] \tag{51}$$

$$\leq 2L\sigma\eta\eta_{\mathbf{w}_f}\,\mathbb{E}\Big[\sqrt{\|\nabla\mathcal{L}_{\text{tot}}^{(s)}(\boldsymbol{\theta}_t^{(S)}) + \Psi_{t-1}\|_2^2} + \sqrt{\|\nabla\mathcal{L}_{\text{focs}}(\boldsymbol{\theta}_{t+1}^{(S)}) + \varepsilon^{(t)}\|_2^2}\Big] \tag{52}$$

$$\leq 2L\sigma\eta\eta_{\mathbf{w}_f}\,\sqrt{\mathbb{E}\big[\|\nabla\mathcal{L}_{\text{tot}}^{(s)}(\boldsymbol{\theta}_t^{(S)}) + \Psi_{t-1}\|_2^2\big] + \mathbb{E}\big[\|\nabla\mathcal{L}_{\text{focs}}(\boldsymbol{\theta}_{t+1}^{(S)}) + \varepsilon^{(t)}\|_2^2\big]} \tag{53}$$

$$\leq 2L\sigma\eta\eta_{\mathbf{w}_f}\,\sqrt{\mathbb{E}\big[\|\nabla\mathcal{L}_{\text{tot}}^{(s)}(\boldsymbol{\theta}_t^{(S)})\|_2^2\big] + \mathbb{E}\big[\|\Psi_{t-1}\|_2^2\big] + \mathbb{E}\big[\|\varepsilon^{(t)}\|_2^2\big] + \mathbb{E}\big[\|\nabla\mathcal{L}_{\text{focs}}(\boldsymbol{\theta}_{t+1}^{(S)})\|_2^2\big]} \tag{54}$$

$$\leq 2L\sigma\eta\eta_{\mathbf{w}_f}\,\sqrt{2\delta^2 + 2\sigma^2} \tag{55}$$

$$\leq 2\sqrt{2}\left(\delta^2 + \sigma^2\right)L\sigma\eta\eta_{\mathbf{w}_f}, \tag{56}$$

where $\eta$ and $\eta_{\mathbf{w}_f}$ are the step sizes for the student parameters $\boldsymbol{\theta}^{(S)}$ and forecaster weights $\mathbf{w}_f$, respectively, and $\delta^2, \sigma^2$ bound the second moments of the corresponding gradients and noise terms.

### C.3  TRAINING DYNAMICS OF INTERLEAVED SCHEDULE: AN EXAMPLE

As shown in Figure 5, In the early phase of the distillation process, forecaster's predictive capability starts at near random. However, forecaster predictions start to stabilize $\geq 60\%$ within the first 1000 minibatches (Total: 7500) of training. The trend is reflected in the left hand side plot, where initially our method performs slightly inferior to Vanilla KD for the initial 1000 steps of the training, but surpasses Vanilla KD thereafter. The post-hoc adjustment to the forecaster outputs is a key aspect in mitigating initial noise to a great extent. We believe that a warm-up phase for initial m% of training steps, where forecaster weights are overridden by $\beta$ can further stabilize early training.

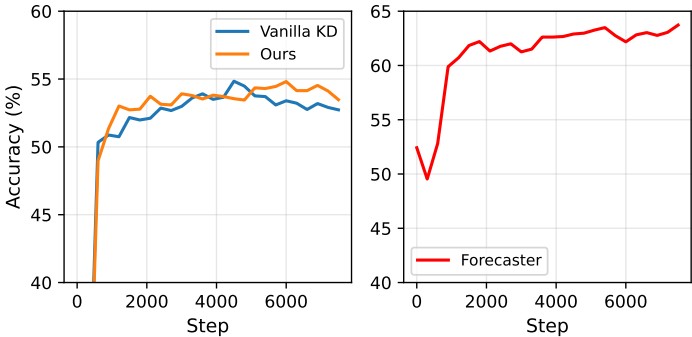

Figure 5: Training dynamics (accuracy plot) for one held-out domain of OfficeHome. The left plot showcases OOD accuracy during a training run, comparing Vanilla KD with our method. Similarly, the right plot showcases forecaster's accuracy on its objective to predict whether the student's prediction is correct.

## D  ADDITIONAL EXPERIMENTAL RESULTS

### D.1  SELECTION OF ROBUST DG ALGORITHM

| ALGORITHM | OFFICEHOME | VLCS | PACS | TERRAINC. | AVG. |
|---|---|---|---|---|---|
| ERM (Vapnik, 1998) | 68.4 | 77.6 | 86.4 | 48.6 | 70.3 |
| GROUPDRO (Sagawa et al., 2020) | 68.6 | 78.0 | 87.0 | 46.7 | 70.1 |
| MIXUP (Yan et al., 2020) | **70.4** | 78.0 | 87.5 | 46.1 | 70.5 |
| MLDG (Li et al., 2017b) | 56.3 | 72.1 | 68.1 | 33.5 | 57.5 |
| CORAL (Sun & Saenko, 2016) | 68.6 | 77.1 | **87.7** | 48.4 | 70.5 |
| MMD (Li et al., 2018a) | 69.2 | 77.5 | 85.6 | **48.7** | 70.3 |
| DANN (Ganin et al., 2016) | 69.7 | **79.7** | 86.3 | 47.4 | **70.8** |
| CDANN (Li et al., 2018b) | 70.0 | 79.0 | 86.0 | 48.3 | **70.8** |

Table 5: Average OOD classification accuracies (%) for all datasets with `ResNet-152`. Model selection: training-domain validation set.

### D.2  DOMAIN-WISE RESULTS ON BENCHMARK DATASETS

Tables $7-14$ report quantitative figures for domain generalization on the OfficeHome (Venkateswara et al., 2017), PACS (Li et al., 2017a), VLCS (Fang et al., 2013), and TerraIncognita (Beery et al., 2018) datasets. We use the `ResNet-18` as the primary network in all experiments. Tables $7-10$ use `ResNet-152` as the teacher network, while Tables $11-14$ use `ViT-L/16` as the teacher network. Each column in the table represents the held-out domain.

### D.3  SENSITIVITY ANALYSIS ON HYPERPARAMETERS $\mu*, \varsigma*, T_s, T_f$ AND $\beta$

Tables $15-17$ report quantitative figures for domain generalization on the OfficeHome Dataset as a function of various configurations of hyperparameters associated to our method: (1) forecaster

| ALGORITHM | OFFICEHOME | VLCS | PACS | TERRAINC. | AVG. |
|---|---|---|---|---|---|
| ERM (Vapnik, 1998) | 71.6 | 78.4 | 83.9 | 38.8 | 68.2 |
| GROUPDRO (Sagawa et al., 2020) | 71.5 | 77.3 | 80.3 | 33.7 | 65.7 |
| MIXUP (Yan et al., 2020) | **72.2** | 76.7 | 81.6 | 42.5 | 68.3 |
| MLDG (Li et al., 2017b) | 70.5 | 76.3 | 81.3 | 40.0 | 67.0 |
| CORAL (Sun & Saenko, 2016) | 71.9 | 78.2 | **84.4** | 40.8 | 68.8 |
| MMD (Li et al., 2018a) | 71.3 | 77.5 | 81.9 | 38.8 | 67.4 |
| DANN (Ganin et al., 2016) | 72.1 | **78.7** | 84.3 | **43.1** | **69.6** |
| CDANN (Li et al., 2018b) | 71.4 | 77.2 | 78.6 | 42.9 | 67.5 |

Table 6: Average OOD classification accuracies (%) for all datasets with `ViT-L/16`. Model selection: training-domain validation set.

| METHOD | A | C | P | R | AVG. |
|---|---|---|---|---|---|
| ERM (Vapnik, 1998) | 50.4 | 48.3 | 65.4 | 69.4 | 58.4 |
| MIXUP (Yan et al., 2020) | 51.5 | 48.6 | 70.3 | 70.2 | 60.2 |
| KD (Hinton et al., 2015b) | 52.7 | 49.7 | **72.1** | 74.2 | 62.2 |
| KD$_{+F+ADJ.}$ | 54.0 | **51.3** | 71.6 | **75.1** | **63.0** |

Table 7: OOD classification accuracy (%) on the OfficeHome dataset. Model selection: training-domain validation set. KD experiments use a `ResNet-152` teacher network.

mean ($\mu^*$) (2) forecaster std ($\varsigma^*$) and (3) interleaved minibatch schedule ($T_s, T_f$), in isolation. Additionally, Table 18 reports the sensitivity in OOD performance on OfficeHome with change in vanilla KD loss coefficient ($\beta$).

### D.4 DOMAIN-WISE RESULTS ON COLOREDMNIST

Table 19 reports quantitative figures on ColoredMNIST (Arjovsky et al., 2020). We include ColoredMNIST to evaluate robustness under correlation shift. These results demonstrate that our proposed adaptive KD method is not limited to benchmarks where domain shifts alter the image distribution, and not the underlying image to label mapping. It also extends to benchmarks like ColoredMNIST where domain changes influence the image to label mapping by introducing spurious correlations between color and label. Here, OOD generalization is challenging for DG algorithms as they can overfit to the spurious correlations (Gulrajani & Lopez-Paz, 2020).

### D.5 DOMAIN GENERALIZATION ON LARGER STUDENT NETWORK

Table 20 presents OOD accuracies on OfficeHome with a `ResNet-50` student. As the student capacity increases, the gap in performance of vanilla KD and teacher network narrows down. This makes the average performance gain achieved with adaptive KD relatively smaller as compared to a small-capacity student.

| METHOD | A | C | P | S | AVG. |
|---|---|---|---|---|---|
| ERM (Vapnik, 1998) | 77.2 | 73.3 | 94.7 | 73.2 | 79.6 |
| CORAL (Sun & Saenko, 2016) | 77.3 | 73.9 | 94.7 | 74.5 | 80.1 |
| KD (Hinton et al., 2015b) | 82.4 | 76.4 | 95.0 | 75.9 | 82.4 |
| KD$_{+F+ADJ.}$ | 82.4 | **77.0** | **95.7** | **76.0** | **82.8** |

Table 8: OOD classification accuracy (%) on the PACS dataset. Model selection: training-domain validation set. KD experiments use a `ResNet-152` teacher network.

| METHOD | C | S | L | V | AVG. |
|---|---|---|---|---|---|
| ERM (Vapnik, 1998) | 96.6 | 58.5 | 65.4 | 65.9 | 71.6 |
| DANN (Ganin et al., 2016) | 96.9 | 58.8 | 65.7 | 66.2 | 71.9 |
| KD (Hinton et al., 2015b) | 97.2 | 60.0 | 67.9 | 68.8 | 73.5 |
| KD$_{+F+ADJ.}$ | **98.3** | **61.3** | **69.8** | **69.2** | **74.7** |

Table 9: OOD classification accuracy (%) on the VLCS dataset. Model selection: training-domain validation set. KD experiments use a `ResNet-152` teacher network.

| METHOD | L100 | L38 | L43 | L46 | AVG. |
|---|---|---|---|---|---|
| ERM (Vapnik, 1998) | 53.2 | 33.0 | 51.4 | **35.9** | 43.4 |
| MMD (Li et al., 2018a) | 47.4 | **37.4** | 50.0 | 35.5 | 42.6 |
| KD (Hinton et al., 2015b) | 50.3 | 36.8 | 52.7 | 34.6 | 43.6 |
| KD$_{+F+ADJ.}$ | **56.7** | 33.6 | **54.4** | 35.5 | **45.1** |

Table 10: OOD classification accuracy (%) on the TerraIncognita dataset. Model selection: training-domain validation set. KD experiments use a `ResNet-152` teacher network.

| METHOD | A | C | P | R | AVG. |
|---|---|---|---|---|---|
| ERM (Vapnik, 1998) | 50.4 | 48.3 | 65.4 | 69.4 | 58.4 |
| MIXUP (Yan et al., 2020) | 51.5 | 48.6 | 70.3 | 70.2 | 60.2 |
| KD (Hinton et al., 2015b) | 55.4 | **53.0** | **72.6** | **74.0** | 63.7 |
| KD$_{+F+ADJ.}$ | **56.3** | 52.6 | 72.5 | 73.2 | 63.7 |

Table 11: OOD classification accuracy (%) on the OfficeHome dataset. Model selection: training-domain validation set. KD experiments use a `ViT-L/16` teacher network.

| METHOD | A | C | P | S | AVG. |
|---|---|---|---|---|---|
| ERM (Vapnik, 1998) | 77.2 | 73.3 | 94.7 | 73.2 | 79.6 |
| CORAL (Sun & Saenko, 2016) | 77.3 | 73.9 | 94.7 | 74.5 | 80.1 |
| KD (Hinton et al., 2015b) | **79.7** | **75.6** | 95.9 | 74.6 | **81.4** |
| KD$_{+F+ADJ.}$ | 78.3 | 74.7 | 95.9 | **75.0** | 81.0 |

Table 12: OOD classification accuracy (%) on the PACS dataset. Model selection: training-domain validation set. KD experiments use a `ViT-L/16` teacher network.

| METHOD | C | S | L | V | AVG. |
|---|---|---|---|---|---|
| ERM (Vapnik, 1998) | 96.6 | 58.5 | 65.4 | 65.9 | 71.6 |
| DANN (Ganin et al., 2016) | **96.9** | 58.8 | 65.7 | 66.2 | 71.9 |
| KD (Hinton et al., 2015b) | 95.9 | 61.5 | 69.7 | 73.6 | 75.2 |
| KD$_{+F+ADJ.}$ | 96.1 | **62.2** | **71.2** | **74.8** | **76.1** |

Table 13: OOD classification accuracy (%) on the VLCS dataset. Model selection: training-domain validation set. KD experiments use a `ViT-L/16` teacher network.

| METHOD | L100 | L38 | L43 | L46 | AVG. |
|---|---|---|---|---|---|
| ERM (Vapnik, 1998) | 53.2 | 33.0 | 51.4 | **35.9** | 43.4 |
| MMD (Li et al., 2018a) | 47.4 | **37.4** | 50.0 | 35.5 | 42.6 |
| KD (Hinton et al., 2015b) | 47.4 | 32.1 | 50.2 | 32.7 | 40.6 |
| OURS | **55.7** | 36.8 | **52.4** | 34.2 | **44.7** |

Table 14: OOD classification accuracy (%) on the TerraIncognita dataset. Model selection: training-domain validation set. KD experiments use a `ViT-L/16` teacher network.

| $(T_s, T_f)$ | A | C | P | R | AVG. |
|---|---|---|---|---|---|
| (1, 1) | **54.0** | 51.3 | 71.6 | **75.1** | **63.0** |
| (1, 10) | 53.4 | 51.2 | **71.8** | 72.7 | 62.3 |
| (10, 1) | 53.0 | 51.5 | 71.3 | 72.7 | 62.2 |
| (10, 10) | 52.3 | **52.4** | 71.3 | 71.9 | 62.0 |

Table 15: OOD classification accuracy (%) on the OfficeHome dataset as a function of the interleaved training schedule. Here, $T_s$ refers to the number of minibatches the student network is trained for, whereas $T_f$ corresponds to the number of meta-minibatches, the forecaster is trained for in the interleaved process.

| $\mu^*$ | A | C | P | R | AVG. |
|---|---|---|---|---|---|
| 0.1 | 52.2 | **51.8** | 69.9 | 71.9 | 61.5 |
| 0.3 | 53.6 | 51.6 | 70.7 | 72.3 | 62.1 |
| 0.5 | **54.0** | 51.3 | **71.6** | **75.1** | **63.0** |
| 0.7 | 52.9 | 51.1 | **71.6** | 72.2 | 62.0 |
| 0.9 | 53.0 | 51.0 | 71.3 | 72.1 | 61.9 |

Table 16: OOD classification accuracy (%) on the OfficeHome dataset as a function of the adjusted forecaster batch mean ($\mu^*$). Here, the standard deviation of adjusted forecaster logits, $\varsigma^* = 0.1$.

| $\varsigma^*$ | A | C | P | R | AVG. |
|---|---|---|---|---|---|
| 0.01 | 52.5 | 51.5 | 71.5 | 72.4 | 62.0 |
| 0.1 | 54.0 | 51.3 | 71.6 | **75.1** | **63.0** |
| 0.5 | 53.6 | **52.0** | 71.9 | 72.3 | 62.5 |
| 1.0 | **54.7** | 51.2 | **72.0** | 73.0 | 62.8 |

Table 17: OOD classification accuracy (%) on the OfficeHome dataset as a function of the adjusted forecaster batch std ($\varsigma^*$). For this analysis, the mean of adjusted forecaster logits, $\mu^* = 0.5$.

| $\mu^*$ | A | C | P | R | Avg. |
|---|---|---|---|---|---|
| 0.1 | **53.6** | **52.5** | 69.8 | 72.2 | 62.1 |
| 0.3 | 53.3 | 51.8 | 71.3 | 72.5 | 62.2 |
| 0.5 | 52.7 | 49.7 | 72.1 | **74.2** | 62.2 |
| 0.7 | 53.5 | 51.7 | 71.6 | 72.5 | 62.3 |
| 0.9 | 53.3 | 51.6 | **72.4** | 72.3 | **62.4** |

Table 18: OOD classification accuracy (%) in Vanilla KD on the OfficeHome dataset as a function of KD loss weight ($\beta$).

| Method | +90% | +80% | -90% | Avg. |
|---|---|---|---|---|
| Teacher | 73.0 | 74.4 | 10.2 | 52.5 |
| ERM (Vapnik, 1998) | 71.6 | 72.9 | 9.9 | 51.5 |
| KD (Hinton et al., 2015b) | **72.7** | 72.2 | 10.1 | 51.6 |
| Ours | 72.5 | **73.8** | **10.3** | **52.2** |

Table 19: OOD classification accuracy (%) on the ColoredMNIST dataset. Model selection: training-domain validation set. Model Architecture: `CNN`.

| Method | A | C | P | R | Avg. |
|---|---|---|---|---|---|
| ERM (Vapnik, 1998) | 59.0 | 49.9 | 73.2 | 75.4 | 64.4 |
| Mixup (Yan et al., 2020) | 61.6 | 54.0 | 75.0 | 75.8 | 66.6 |
| KD (Hinton et al., 2015b) | **64.6** | 55.2 | 75.7 | 77.5 | 68.3 |
| KD$_{+F+Adj.}$ | 62.0 | **56.9** | **76.7** | **78.8** | **68.6** |

Table 20: OOD classification accuracy (%) on the OfficeHome dataset with a `ResNet-50` student. Model selection: training-domain validation set. KD experiments use a `ResNet-152` teacher network.

