# OpenReview forum: "Early Layer Readouts for Robust Knowledge Distillation"
_ICLR.cc/2026/Conference — ICLR 2026 Conference Withdrawn Submission_

### Official Review · Reviewer_XXf7 · 2025-10-15

**Soundness:** 2
**Presentation:** 2
**Contribution:** 2
**Rating:** 2
**Confidence:** 3

**Summary:**

This paper proposes a KD–based training strategy for OOD generalization. The authors first argue that training compact student models via simple KD from a teacher with strong OOD performance can often surpass standalone algorithmic DG methods. They further note that prior OOD-oriented KD approaches predominantly focus on the teacher’s design or the teacher–student relationship, leaving the design of the student model underexplored. To address this, the authors introduce a forecaster that quantifies per-sample difficulty using auxiliary models built on the student’s internal representations together with uncertainty measures. The KD loss is then reweighted on a per-sample basis according to the predicted difficulty. Experiments on four DomainBed datasets with ResNet-18 demonstrate the effectiveness of the proposed approach.

**Strengths:**

- The paper highlights a student-model training framework for KD in the context of OOD generalization, a direction that has been relatively underexplored.
- It proposes per-sample weighting via a meta-model that leverages the student’s internal representations and uncertainty measures.

**Weaknesses:**

- Unclear linkage between motivation and design: The paper starts from the observation that KD is effective for OOD but uniform weighting is brittle, yet the causal chain that justifies why early-layer readouts plus uncertainty lead to instance weighting as the essential solution remains vague. For each design choice (using early student layers; adopting entropy and margin; alternating training; applying post-hoc adjustment), the paper should clarify why it is fundamentally or theoretically effective under OOD conditions.
- Instability of alternating training: The student, auxiliary models, and the forecaster are all trained progressively from the beginning. Naturally, the auxiliary model’s correctness on the student and the forecaster’s weight predictions start near random in the early phase. The paper does not deeply examine how this phase affects student learning, nor when the auxiliary/forecaster models begin to exhibit reliable behavior in terms of correctness and weight prediction.
- Limited experimental scope: Validation is restricted to a small set of datasets within the DomainBed framework. The paper does not evaluate on more challenging datasets such as DomainNet, nor on correlation-shift settings (e.g., Colored MNIST) beyond diversity shift, and it does not report results across multiple seeds.
- Limited baselines: The authors adopt only training-algorithm baselines for KD, but many OOD-focused methods that leverage weight averaging also exist such as DiWA[1], and these are known to be more effective than designing new training algorithms.
- Missing sensitivity analysis for newly introduced hyperparameters: The two hyperparameters introduced to adjust the forecaster’s predictions, $\varsigma^\star$ and $\mu^\star$, appear to be fixed to 0.1 and 0.5, respectively, without justification. The paper should explain how they were chosen and provide a sensitivity analysis.
- Typos and minor presentation issues: line 037 “treates,” line 038 “exisiting,” line 059 “consitently,” and line 260 likely should use “.” rather than “:”.

[1] Rame, Alexandre, et al. "Diverse weight averaging for out-of-distribution generalization." Advances in Neural Information Processing Systems 35 (2022): 10821-10836.

**Questions:**

See weaknesses

---

> ### Author Response · Authors · 2025-11-25
> **Response to Reviewer XXf7 (1/3)**
>
> We thank the reviewer for their valuable feedback on our submission. We appreciate the reviewer's acknowledgement of the novelty in exploring student-centric KD for OOD. We address the questions raised by the reviewer as follows:
>
> *Unclear linkage between motivation and design: The paper starts from the observation that KD is effective for OOD but uniform weighting is brittle, yet the causal chain that justifies why early-layer readouts plus uncertainty lead to instance weighting as the essential solution remains vague. For each design choice (using early student layers; adopting entropy and margin; alternating training; applying post-hoc adjustment), the paper should clarify why it is fundamentally or theoretically effective under OOD conditions.*
>
> We now provide theoretical support for why early-layer readout signals serve as an effective proxy for sample difficulty and how they relate to reducing domain generalization error in the Supplementary Sections B and C in the revised draft of the paper. At a high level, early-layer signals capture feature variations across domains and yield an upper bound on the worst-case domain generalization error. While prior ERM-based works primarily analyze feature variation using only last-layer representations, our analysis offers a more grounded perspective by showing how signals from multiple layers contribute to cross-domain feature variation.
>
> ---
>
> *Limited experimental scope: Validation is restricted to a small set of datasets within the DomainBed framework. The paper does not evaluate on more challenging datasets such as DomainNet, nor on correlation-shift settings (e.g., Colored MNIST) beyond diversity shift, and it does not report results across multiple seeds.*
>
> We omit DomainNet from our evaluation due to compute constraints. However, our additional experiments on a transformer-based text model showcases that our method generalizes across modality and architecture. Specifically, we choose two representative ID - OOD task pairs following the splits proposed in GLUE-X [1] in our evaluation: (1) MPRC - QQP (Task: Paraphrase) (2) MNLI - MNLImis (Task: Natural Language Inference). For the student model, we use a 4-layer compact BERT network utilizing layers 0 to 2 as early exits, each with a lightweight single layer linear auxiliary head. We use a 2-layer feed-forward network with ReLU for the forecaster network.
>
> | Method                              | GLUE (MPRC → QQP) | MNLI (Train MNLI → Mismatched) |
> |-------------------------------------|--------------------|--------------------------------|
> | **Teacher (bert-base-uncased)**     | 65.73              | 84.34                          |
> | **ERM**                             | 57.11              | 74.57                          |
> | **KD**                              | 58.42              | 74.83                          |
> | **Ours**                            | 60.90              | 75.40                          |
>
>
> As shown, our method yields an average increase of 1.53% relative to the standard KD in the text modality with a transformer-based architecture, showcasing the generality of our work across modality and model architecture.
>
> While we do not report results averaged across multiple seeds due to compute constraints, we follow the default hyperparameter and seed setup from DomainBed [2] allowing reproducibility and fair comparison to the baselines.
>
> [1] Li et al, 2022,  GLUE-X: Evaluating Natural Language Understanding Models from an Out-of-distribution Generalization Perspective
>
> [2] Ishaan Gulrajani and David Lopez-Paz. In search of lost domain generalization, 2020
>
> ---
>
> *Typos and minor presentation issues: line 037 “treates,” line 038 “exisiting,” line 059 “consitently,” and line 260 likely should use “.” rather than “:”.*
>
> We have incorporated suggestions on minor presentation issues in the revised version of the paper.
>
> ---

---

> ### Author Response · Authors · 2025-11-25
> **Response to Reviewer XXf7 (2/3)**
>
> *Limited baselines: The authors adopt only training-algorithm baselines for KD, but many OOD-focused methods that leverage weight averaging also exist such as DiWA[1], and these are known to be more effective than designing new training algorithms.*
>
> DiWA proposes the idea of merging weights from multiple independent model runs on a dataset with different hyperparameter configurations suggesting an ensemble of functionally diverse individual models in the parameter space to effectively reduce generalization error. Particularly DiWA trains N models independently using a base algorithm like ERM with multiple trials of hyperparameter search on each model and then merge best-performing model checkpoints to obtain a robust merged model. By design, DiWA is simpler, but computationally more involved due to the many training runs and post-hoc averaging for each dataset. Compared to leveraging KD for DG as an algorithm, DiWA is a post-training strategy which could leverage any base algorithm for post-hoc merging. We believe DiWA is orthogonal as a baseline in our current setup. However, DiWA can be applied as a complementary strategy on functionally diverse distilled student networks for additive improvements for DG. Although we do not explore this direction here due to compute constraints and rebuttal timelines, we acknowledge the possible usage of KD to learn functionally diverse distilled student networks and subsequently leveraging DiWA for additive improvements on DG as an avenue for future work.
>
> ---
>
> *Missing sensitivity analysis for newly introduced hyperparameters: The two hyperparameters introduced to adjust the forecaster’s predictions, $\varsigma^\star$ and $\mu^\star$, appear to be fixed to 0.1 and 0.5, respectively, without justification. The paper should explain how they were chosen and provide a sensitivity analysis.*
>
> We chose $\mu^\ast$ and $\zeta^\ast$ to be 0.5 and 0.1 as pragmatic, dataset-agnostic hyperparameters. $\mu^\ast = 0.5$ centers the post-hoc adjustment to allow equal contribution from teacher soft predictions and ground-truth supervision. This allows the training signal to not be overly dominated by the teacher or the ground truth. $\zeta^\ast = 0.1$ allows a controlled spread around the center, enabling an instance-based adjustment proposed by the forecaster without a strong drift in either direction. We conduct sensitivity analysis by (1) varying $\mu^\ast$ from 0.1 to 0.9 while keeping $\zeta^\ast$ fixed at 0.1 and (2) varying $\zeta^\ast$ from 0.01 to 1.0 while keeping $\mu^\ast$ fixed at 0.5. Our analysis shows that OOD performance does not degrade sharply across various choices of $(\mu^\ast, \zeta^\ast)$ pairs. While the hyperparameters can be tuned per-dataset or per-domain within a dataset, we recommend the practical default choices under compute constraints.
>
>
>
> **Sensitivity analysis on $\mu^{\ast}$ with $\zeta^{\ast}$=0.1**
>
>
> | $\mu^{\ast}$  | A    | C    | P    | R    | Avg.  |
> |-----|------|------|------|------|-------|
> | 0.1 | 52.2 | 51.8 | 69.9 | 71.9 | 61.45 |
> | 0.3 | 53.6 | 51.6 | 70.7 | 72.3 | 62.05 |
> | 0.5 | 54.0 | 51.3 | 71.6 | 75.1 | 63.00 |
> | 0.7 | 52.9 | 51.1 | 71.6 | 72.2 | 61.95 |
> | 0.9 | 53.0 | 51.0 | 71.3 | 72.1 | 61.85 |
>
>
> **Sensitivity analysis on $\zeta^{\ast}$ with $\mu^\ast = 0.5$**
>
>
> | $\zeta^{\ast}$   | A    | C    | P    | R    | Avg.    |
> |------|------|------|------|------|---------|
> | 0.01 | 52.5 | 51.5 | 71.5 | 72.4 | 61.975  |
> | 0.1  | 54.0 | 51.3 | 71.6 | 75.1 | 63.000  |
> | 0.5  | 53.6 | 52.0 | 71.9 | 72.3 | 62.450  |
> | 1.0  | 54.7 | 51.2 | 72.0 | 73.0 | 62.725  |
>
> Sensitivity analysis on $T_s$  and $T_f$:
>
> Similarly, we fix $T_s$ = 1 and $T_f$ = 1 in our experiments. With $T_s$  = 1 and $T_f$  = 1, we simply ensure that the forecaster is updated as frequently as the student, preventing the forecaster weights becoming stale as the student representation evolves during training. To assess how the frequency of updates to the forecaster affect the downstream OOD performance of the student network, we conduct a sensitivity analysis by varying  $T_s$  and $T_f$ across several configurations shown below. As shown, we observe a small but consistent drop in average OOD performance in these alternative configurations. These observations suggest that the forecaster network is negatively affected in both cases of (1) infrequent updates (10; 1) (lagging behind current student configuration) and too frequent updates (10; 10 / 1; 10) (overfitted to current student configuration).
>
> | ($T_s$, $T_f$) | A    | C    | P    | R    | Avg.    |
> |----------|------|------|------|------|---------|
> | (1; 1)   | 54.0 | 51.3 | 71.6 | 75.1 | 63.000  |
> | (10; 1)  | 53.0 | 51.5 | 71.3 | 73.1 | 62.225  |
> | (1; 10)  | 53.4 | 51.2 | 71.8 | 72.7 | 62.275  |
> | (10; 10) | 52.3 | 52.4 | 71.3 | 71.9 | 61.975  |
>
> ---

---

> ### Author Response · Authors · 2025-11-26
> **Response to Reviewer XXf7 (3/3)**
>
> *Instability of alternating training: The student, auxiliary models, and the forecaster are all trained progressively from the beginning. Naturally, the auxiliary model’s correctness on the student and the forecaster’s weight predictions start near random in the early phase. The paper does not deeply examine how this phase affects student learning, nor when the auxiliary/forecaster models begin to exhibit reliable behavior in terms of correctness and weight prediction.*
>
> Figure 5 in Supplementary (Appendix) C.3 illustrates the training dynamics for one held-out domain of OfficeHome. The right plot showcases forecaster's accuracy on its objective to predict whether the student's prediction is correct. While it starts near random, it starts to stabilize above within 1000 minibatches (Total: 7500) of training. The trend is reflected in the left hand side plot, where initially our method performs slightly inferior to Vanilla KD for the initial 1000 steps but improves over it thereafter. The post-hoc adjustment to the forecaster outputs is a key aspect in mitigating initial noise to a great extent. We believe that a warm-up phase for initial m% of training steps, where forecaster weights are overridden by \beta can further stabilize early training phase.
>
> ---
>
> We thank the reviewer for their valuable suggestions and questions, which have helped us to strengthen our work with additional experiments and clarifications. The revised draft of the paper incorporates the entirety of the discussion to address all review concerns and highlight the extended merits of our work. We kindly request a reconsideration of the current score and an appropriate increase to better reflect the contributions of our work.

---

> > ### Comment · Reviewer_XXf7 · 2025-11-27
> > **Reply to authors**
> >
> > Thank you for addressing my concerns.
> >
> > The additions on theoretical grounding, the early-phase instability under alternating training, and the hyperparameter sensitivity analyses have satisfactorily resolved those points.
> >
> > However, I still have substantial reservations about the generality and scalability of the experiments. While I appreciate the stated constraints on compute and time, the current empirical scope does not sufficiently justify the proposed method in broader settings.
> >
> > I have updated my score accordingly, but for the reasons above, I still do not consider the paper to be at the accept line.

---

> > > ### Author Response · Authors · 2025-12-03
> > > **Response to Reviewer XXf7 on Generality and Scalability**
> > >
> > > In addition to BERT based text classification experiments, we now include results on ColoredMNIST to evaluate robustness under correlation shift. These results demonstrate that our proposed adaptive KD method is not limited to benchmarks where domain shifts alter the image distribution, and not the underlying image to label mapping. It also extends to benchmarks like ColoredMNIST where domain changes influence the image to label mapping by introducing spurious correlations between color and label. As stated in [1], this is challenging for algorithms as they can overfit to the spurious correlations.
> > >
> > > | Method  | +90%   | +80% | -90%  | Avg. |
> > > |---------|-------|-------|-------|--------|
> > > | Teacher | 73.0  | 74.4  | 10.2  | 52.5   |
> > > | ERM (DB. SotA)     | 71.6  | 72.9  | 9.9   | 51.5   |
> > > | KD      | **72.7**  | 72.2  | 10.1  | 51.6   |
> > > | Ours | 72.5  | **73.8**  | **10.3**  | **52.2**   |
> > >
> > > In the above experiments, the student follows the CNN based network hand-tuned for MNIST [1] , whereas, the teacher is a deeper variant based on the same design.
> > >
> > > In addition to generality, we also present the OOD robustness accuracies on OfficeHome using a larger student network (ResNet50) with the same ResNet152 Teacher. These results confirm that our approach improves over vanilla KD even with larger student capacity, highlighting the scalability of the approach beyond a small-student (ResNet18). However, it is important to note that as the student capacity increases, the gap in performance of vanilla KD and teacher network narrows down. This makes the average performance gain achieved with adaptive KD relatively smaller as compared to a small-capacity student.
> > >
> > > | Method |   A   |   C   |   P   |   R   | Avg.  |
> > > |--------|-------|-------|-------|-------|-------|
> > > | ERM    | 59.0  | 49.9  | 73.2  | 75.4  | 64.4  |
> > > | Mixup (DB. SotA)  | 61.6  | 54.0  | 75.0  | 75.8  | 66.6  |
> > > | KD     | **64.6**  | 55.2  | 75.7  | 77.5  | 68.3  |
> > > | Ours   | 62.0  | **56.9**  | **76.7**  | **78.8**  | **68.6**  |
> > >
> > > [1] Ishaan Gulrajani and David Lopez-Paz. In search of lost domain generalization, 2020

---

### Official Review · Reviewer_JkxR · 2025-10-30

**Soundness:** 2
**Presentation:** 1
**Contribution:** 2
**Rating:** 2
**Confidence:** 5

**Summary:**

The paper proposes an adaptive KD framework for domain generalization where a lightweight forecaster uses early-layer readouts (auxiliary heads) and uncertainty features (entropy, confidence margin) to reweight per-instance contributions of supervised loss vs. teacher KL during student training. The forecaster is trained interleaved with the student and discarded at inference, so deployment cost matches vanilla KD.

**Strengths:**

1. Clear motivation & pragmatic aim. KD already provides strong OOD gains; adaptivity addresses per-sample variability and teacher bias without incurring inference overhead.
2. Student-centric design. Focuses on when and how the student should trust the teacher, rather than only on teacher improvements; leverages early readouts as signals.

**Weaknesses:**

1. The method feels incremental relative to prior KDDG work [1], offering an early-readout reweighting tweak rather than a clearly novel principle.
2. Magnitude of gains is modest. Improvements over a strong KD baseline average about +1.0–1.2% and vary by dataset; some cells are unchanged (TEACHER NETWORK: ViT-L/16). It’s not always clear whether the effect size justifies a new module vs. careful KD tuning.
3. The paper lacks a systematic hyperparameter sensitivity analysis (e.g., τ, α/β, μ*/σ*, interleaving schedule), leaving the robustness of the reported gains unclear.
4. The evaluation uses few and relatively small benchmarks and omits larger DomainBed suites like DomainNet, limiting evidence for scalability.
5. The comparison set is outdated, lacking strong recent KD baselines (e.g., RISE [2], BOLD [3]), which weakens the competitiveness claim.
6. The paper offers only empirical validation and lacks theoretical grounding to explain why early-layer readout–driven reweighting should improve OOD generalization.
7. Experiments use only a small student (ResNet-18) distilled from a very strong teacher (ViT-L/16); the authors should evaluate larger students (e.g., ResNet-50, ViT-B) to test whether gains persist—especially since improvements with ResNet-18 are already modest.

[1] 2021 - ACM MM - Embracing the Dark Knowledge: Domain Generalization Using Regularized Knowledge Distillation

[2] 2023 - ICCV - A Sentence Speaks a Thousand Images: Domain Generalization through Distilling CLIP with Language Guidance

[3] 2025 - IJCAI - Balancing Invariant and Specific Knowledge for Domain Generalization with Online Knowledge Distillation

**Questions:**

See Weakness.

---

> ### Author Response · Authors · 2025-11-25
> **Response to Reviewer JkxR 1/4**
>
> We thank the reviewer for their valuable feedback and appreciate their positive remarks regarding the paper’s clear motivation, practical aims, and its contributions in addressing per-sample variability and teacher bias without added inference overhead.
>
> We respond to the questions raised by the reviewer as follows:
>
> > The method feels incremental relative to prior KDDG work [1], offering an early-readout reweighting tweak rather than a clearly novel principle.
>
> While both KDDG [1] and our method leverage knowledge distillation to improve domain generalization, we believe the claim of our work being incremental oversimplifies the novel technical contributions of our work. KDDG utilizes a rule-based static thresholding mechanism (referred to as gradient filter) based on student’s prediction confidence to downweigh easy samples. The heuristic in KDDG is handcrafted and fixed. On the other hand, we adopt a data-driven, batch-aware weighing function (forecaster), leveraging early layer uncertainty signals to determine sample difficulty. While KDDG uses final-layer confidence of the student network, the forecaster processes information from all early ($\ell = 1$) to the penultimate layer ($\ell = L - 1$) of the student network, without requiring the need for layer selection. This allows the forecaster to capture inter-layer nuances in effectively gauging sample ambiguity which the KDDG formulation does not allow.
>
> Formally, the KDDG weighing function is formulated as,
>
> $$
> \begin{aligned}
> f(\omega) &= \omega, && p \le \eta,
> f(\omega) &= \left( \frac{\eta + 1 - 2p}{1 - \eta} \right)^2 \omega, && \eta < p \le \frac{1+\eta}{2},
> f(\omega) &= 0, && p > \frac{1+\eta}{2}.
> \end{aligned}
> $$
>
> where $p$ is the student’s confidence (final layer max. softmax probability) and $\omega$ denotes the gradient of the student model.
>
> This weight is applied to both the cross-entropy and the KD loss.
>
> In our method, the weightage is as follows
>
> $w_{\psi_f}^{\text{adj}} = \sigma(\varsigma^{\ast}(\frac{\boldsymbol{z_f} - \mu_{B}}{\varsigma_{B}})+ \mu^{\ast})$
>
> where $z_f$ are the forecaster prediction logits, adjusted to center around $\mu^\ast$ with spread $\varsigma^\ast$.
>
>
> Lastly, there is no publicly available official codebase for KDDG, to the best of our knowledge, to include it as a potential baseline.
>
>
> --------------
>
> > The comparison set is outdated, lacking strong recent KD baselines (e.g., RISE [2], BOLD [3]), which weakens the competitiveness claim.
>
> RISE[2] leverages CLIP as teacher, utilizing both vision and language as supervision in the distillation process. RISE tries to regularize student’s learned features to be closer to semantic text embedding space, as the text space provides more domain-agnostic representations.
> BOLD[3] focuses on leveraging a CLIP backbone (frozen; domain invariant) with domain-expert adapter networks (trainable) as sources of teacher supervision with dynamic balancing between invariant and domain-specific gradients. Along with the student network, the domain-specific teacher experts are trained online during the distillation process using reverse KL from the student and ground-truth supervision.
> We believe that the abovementioned methodologies operate under assumptions that differ fundamentally from our setting: (1) A teacher-centric design choice with usage of cross modal regularization in RISE, (2) BOLD focuses on online KD and multi-expert regularization (domain-experts and invariant CLIP backbone). These assumptions make them orthogonal to our work rather than being directly comparable baselines. However, we do acknowledge these works as potential complementary design choices that could potentially be integrated along with our student-centric approach. While we do not take this direction during the rebuttal phase, owing to time and resource constraints, we pose this as a future work in adaptive KD for Domain Generalization.
>
> [1] 2021 - ACM MM - Embracing the Dark Knowledge: Domain Generalization Using Regularized Knowledge Distillation
>
> [2] 2023 - ICCV - A Sentence Speaks a Thousand Images: Domain Generalization through Distilling CLIP with Language Guidance
>
> [3] 2025 - IJCAI - Balancing Invariant and Specific Knowledge for Domain Generalization with Online Knowledge Distillation

---

> ### Author Response · Authors · 2025-11-25
> **Response to Reviewer JkxR 2/4**
>
> > Sensitivity Analysis on Hyperparameters $\mu^\ast$ $\zeta^\ast$, $T_s$, $T_f$
>
> We chose $\mu^{\ast}$ and $\zeta^{\ast}$ to be 0.5 and 0.1 as pragmatic, dataset agnostic hyperparameters. $\mu^{\ast}$= 0.5 centers the post-hoc adjustment to allow equal contribution from teacher soft predictions and ground-truth supervision. This allows the training signal to not be overly dominated by the teacher or the ground-truth. $\zeta^{\ast}$ = 0.1 allows a controlled spread around the center, allowing an instance-based adjustment proposed by the forecaster without a strong drift in either direction. We conduct sensitivity analysis by (1) varying $\mu^{\ast}$ from 0.1 to 0.9 while keeping $\zeta^{\ast}$ fixed to 0.1 and (2) varying $\zeta^{\ast}$ is varied from 0.01 to 1.0 while keeping $\mu^{\ast}$ fixed to 0.5. Our analysis shows that OOD performance does not degrade sharply across various choices of ($\mu^{\ast}$, $\zeta^{\ast}$) pairs. While the hyperparameters can be tuned per-dataset or per-domain within a dataset, we recommend the practical default choices under compute constraints.
>
> **Sensitivity analysis on $\mu^{\ast}$ with $\zeta^{\ast}$=0.1**
>
>
> | $\mu^{\ast}$  | A    | C    | P    | R    | Avg.  |
> |-----|------|------|------|------|-------|
> | 0.1 | 52.2 | 51.8 | 69.9 | 71.9 | 61.45 |
> | 0.3 | 53.6 | 51.6 | 70.7 | 72.3 | 62.05 |
> | 0.5 | 54.0 | 51.3 | 71.6 | 75.1 | 63.00 |
> | 0.7 | 52.9 | 51.1 | 71.6 | 72.2 | 61.95 |
> | 0.9 | 53.0 | 51.0 | 71.3 | 72.1 | 61.85 |
>
>
> **Sensitivity analysis on $\zeta^{\ast}$ with $\mu^\ast = 0.5$**
>
>
> | $\zeta^{\ast}$   | A    | C    | P    | R    | Avg.    |
> |------|------|------|------|------|---------|
> | 0.01 | 52.5 | 51.5 | 71.5 | 72.4 | 61.975  |
> | 0.1  | 54.0 | 51.3 | 71.6 | 75.1 | 63.000  |
> | 0.5  | 53.6 | 52.0 | 71.9 | 72.3 | 62.450  |
> | 1.0  | 54.7 | 51.2 | 72.0 | 73.0 | 62.725  |
>
>
> ---------------------
>
> Similarly, we fix $T_s$ = 1 and $T_f$ = 1 in our experiments. With $T_s$  = 1 and $T_f$  = 1, we simply ensure that the forecaster is updated as frequently as the student, preventing the forecaster weights becoming stale as the student representation evolves during training. To assess how the frequency of updates to the forecaster affect the downstream OOD performance of the student network, we conduct a sensitivity analysis by varying  $T_s$  and $T_f$ across several configurations shown below. As shown, we observe a small but consistent drop in average OOD performance in these alternative configurations. These observations suggest that the forecaster network is negatively affected in both cases of (1) infrequent updates (10; 1) (lagging behind current student configuration) and too frequent updates (10; 10 / 1; 10) (overfitted to current student configuration).
>
> | ($T_s$, $T_f$) | A    | C    | P    | R    | Avg.    |
> |----------|------|------|------|------|---------|
> | (1; 1)   | 54.0 | 51.3 | 71.6 | 75.1 | 63.000  |
> | (10; 1)  | 53.0 | 51.5 | 71.3 | 73.1 | 62.225  |
> | (1; 10)  | 53.4 | 51.2 | 71.8 | 72.7 | 62.275  |
> | (10; 10) | 52.3 | 52.4 | 71.3 | 71.9 | 61.975  |
>
>
>
> -------------

---

> ### Author Response · Authors · 2025-11-25
> **Response to Reviewer JkxR 3/4**
>
> > The evaluation uses few and relatively small benchmarks and omits larger DomainBed suites like DomainNet, limiting evidence for scalability.
>
> We omit DomainNet from our evaluation due to compute constraints. However, our additional experiments on a transformer-based text model showcases that our method generalizes across modality and architecture. Specifically, we choose two representative ID - OOD task pairs following the splits proposed in [1], in our evaluation: (1) MPRC - QQP (Task: Paraphrase) (2) MNLI - MNLImis (Task: Natural Language Inference). For the student model, we use a 4-layer BERT utilizing layers 0 to 2 as early exits, each with a lightweight single layer linear auxiliary head. We use a 2-layer feed-forward network with ReLU for the forecaster network.
>
> | Method                              | GLUE (MPRC → QQP) | MNLI (Train MNLI → Mismatched) |
> |-------------------------------------|--------------------|--------------------------------|
> | **Teacher (bert-base-uncased)**     | 65.73              | 84.34                          |
> | **ERM**                             | 57.11              | 74.57                          |
> | **KD**                              | 58.42              | 74.83                          |
> | **Ours**                            | 60.90              | 75.40                          |
>
>
> > The paper offers only empirical validation and lacks theoretical grounding to explain why early-layer readout–driven reweighting should improve OOD generalization.
>
> We thank the reviewer for the suggestion. In Section B of the supplementary material of the revised draft, we provide theoretical support for why early-layer readout signals serve as an effective proxy for sample difficulty and how they relate to reducing domain generalization error. At a high level, early-layer signals capture feature variations across domains and yield an upper bound on the worst-case domain generalization error. While prior ERM-based works primarily analyze feature variation using only last-layer representations, our analysis offers a more grounded perspective by showing how signals from multiple layers contribute to cross-domain feature variation. Further, we cast the forecaster and student model training as a bilevel optimization problem and provide in Supplementary section C on convergence guarantees for both forecaster and student model.

---

> ### Author Response · Authors · 2025-11-26
> **Response to Reviewer JkxR 4/4**
>
> > Experiments use only a small student (ResNet-18) distilled from a very strong teacher (ViT-L/16); the authors should evaluate larger students (e.g., ResNet-50, ViT-B) to test whether gains persist—especially since improvements with ResNet-18 are already modest.
>
> We get a larger exploitable performance gap between a very small student (ResNet18) and a large teacher (ViT-L/16, ResNet152), which allows for a controlled setting to showcase KD and our methodology's contribution over the raw capacity of the student network. With larger student networks, this attribution of performance gains to the training algorithm v/s model capacity may become unclear. Moreover, the usage of a smaller student network is also motivated from a practical model deployment under resource constraints perspective, as stated in the paper.
>
> ---
>
> We thank the reviewer for their constructive feedback and questions, which have helped us to strengthen our work with additional experiments and clarifications. The revised draft of the paper incorporates the entirety of the discussion to address all reviewer concerns and highlight the extended merits of our work. We kindly request a reconsideration of the current score and an appropriate increase to better reflect the contributions of our work.

---

### Official Review · Reviewer_NoZD · 2025-10-31

**Soundness:** 3
**Presentation:** 3
**Contribution:** 3
**Rating:** 6
**Confidence:** 2

**Summary:**

This paper addresses out-of-distribution (OOD) generalization in knowledge distillation by proposing an adaptive framework that uses early layer predictions to dynamically weight the loss components. The authors introduce a "forecaster" meta-network that leverages auxiliary classifiers at intermediate layers, along with uncertainty measures (entropy and confidence margin), to predict sample difficulty and reweight the balance between supervised loss and distillation loss on a per-instance basis. The method is evaluated on domain generalization benchmarks (OfficeHome, PACS, VLCS, TerraIncognita) and shows consistent improvements over vanilla KD (+1.0-1.2% average accuracy) while adding no inference overhead.

**Strengths:**

Strong empirical validation: Consistent improvements across 4 benchmarks (OfficeHome, PACS, VLCS, TerraIncognita) with both ResNet-152 and ViT-L/16 teachers (Tables 3, 6-13)

No inference overhead: The forecaster and auxiliary networks are discarded after training, maintaining deployment efficiency—a critical practical consideration clearly addressed

Thoughtful design choices: The post-hoc adjustment mechanism is well-motivated by Figure 3, which shows forecaster collapse without normalization. The ablation in Figure 2 demonstrates the importance of uncertainty features (AUC improves from 0.56 to 0.82)

Fair experimental setup: Uses DomainBed codebase, follows established protocols, compares against multiple baselines including best-performing DG algorithms per dataset (Tables 4-5)

Clear identification of vanilla KD's strength: The observation that vanilla KD often outperforms standalone DG algorithms (Section 1, Table 3) is valuable and underexplored in prior work

**Weaknesses:**

Limited architectural generality: The approach is tightly coupled to ResNet's residual block structure (4 auxiliary classifiers at specific stages). Critical concern: No evidence provided that this method generalizes to other architectures (Vision Transformers, EfficientNets, etc.) or modalities beyond image classification. This significantly limits practical applicability.

Modest improvements: Average gains of 1.0-1.2% over vanilla KD are consistent but relatively small. On individual domains, results are mixed (e.g., PACS in Table 11: some domains improve, others decline slightly). The cost-benefit tradeoff of added training complexity may not be compelling for practitioners.

Insufficient theoretical justification: Why should early layer confidence specifically help with OOD generalization rather than just indicating general sample difficulty? The paper treats these as equivalent without rigorous justification. Early layers capture low-level features—the connection to domain shift is unclear.

Ad-hoc forecaster design: The forecaster architecture (1D conv + linear layer, Sections 4.2) appears hand-crafted for ResNet. Hyperparameters for post-hoc adjustment (µ*=0.5, ς*=0.1) are not justified or ablated. Algorithm 1 introduces multiple hyperparameters (Ts, Tf) without sensitivity analysis.

Limited scope: Only evaluated on small-scale image classification benchmarks. No experiments on larger-scale datasets (ImageNet), other modalities (NLP, audio), or other student architectures beyond ResNet-18.
Incomplete analysis:

No comparison with other sample reweighting approaches for KD
Table 1 shows the method without adjustment fails, but is relegated to one dataset
Missing computational cost analysis during training (forecaster adds overhead even if discarded later)

**Questions:**

Same as above.

---

> ### Author Response · Authors · 2025-11-23
> **Response to Reviewer NoZD 1/3**
>
> We thank the reviewer for their constructive assessment. We are glad the motivation for using early-layer signals and uncertainty features to guide adaptive KD was clear. We also appreciate the reviewer;s recognition of evaluation across different benchmarks and across different teacher models. The reviewer’s note that vanilla KD is a strong DG baseline aligns with our motivation, and our method aims to strengthen this behavior without adding inference cost. We will reflect these clarifications in the final version.
>
> ## Limited architectural generality, forecaster design motivations, limited scope.
> The reviewer notes that our method is tightly coupled to ResNet’s residual block architecture. We address the generality concern with experiments evaluating OOD robustness in a transformer-based NLP model, Bert, addressing both cross-architecture and cross-modality generalization. Specifically, we choose two representative ID - OOD task pairs following the splits proposed in Li et al, 2022 https://arxiv.org/abs/2211.08073 in our evaluation: (1) MPRC - QQP (Task: Paraphrase) (2) MNLI - MNLImis (Task: Natural Language Inference). For the student model, we use a 4-layer BERT utilizing layers 0 to 2 as early exits, each with a lightweight single layer linear auxiliary head. We use a 2-layer feed-forward network with ReLU for the forecaster network.
>
> | Method                              | GLUE (MPRC → QQP) | MNLI (Train MNLI → Mismatched) |
> |-------------------------------------|--------------------|--------------------------------|
> | **Teacher (bert-base-uncased)**     | 65.73              | 84.34                          |
> | **ERM**                             | 57.11              | 74.57                          |
> | **KD**                              | 58.42              | 74.83                          |
> | **Ours**                            | 60.90              | 75.40                          |
>
>
> As shown, our method yields an average increase of 1.53 pct relative to the standard KD in the text modality with a transformer-based architecture, showcasing the generality of our work across modality and model architecture.
>
> ## Sensitivity Analysis on Hyperparameters $\mu^\ast$ $\zeta^\ast$, $T_s$, $T_f$
>
> We chose $\mu^{\ast}$ and $\zeta^{\ast}$ to be 0.5 and 0.1 as pragmatic, dataset agnostic hyperparameters. $\mu^{\ast}$= 0.5 centers the post-hoc adjustment to allow equal contribution from teacher soft predictions and ground-truth supervision. This allows the training signal to not be overly dominated by the teacher or the ground-truth. $\zeta^{\ast}$ = 0.1 allows a controlled spread around the center, allowing an instance-based adjustment proposed by the forecaster without a strong drift in either direction. We conduct sensitivity analysis by (1) varying $\mu^{\ast}$ from 0.1 to 0.9 while keeping $\zeta^{\ast}$ fixed to 0.1 and (2) varying $\zeta^{\ast}$ is varied from 0.01 to 1.0 while keeping $\mu^{\ast}$ fixed to 0.5. Our analysis shows that OOD performance does not degrade sharply across various choices of ($\mu^{\ast}$, $\zeta^{\ast}$) pairs. While the hyperparameters can be tuned per-dataset or per-domain within a dataset, we recommend the practical default choices under compute constraints.
>
> **Sensitivity analysis on $\mu^{\ast}$ with $\zeta^{\ast}$=0.1**
>
>
> | $\mu^{\ast}$  | A    | C    | P    | R    | Avg.  |
> |-----|------|------|------|------|-------|
> | 0.1 | 52.2 | 51.8 | 69.9 | 71.9 | 61.45 |
> | 0.3 | 53.6 | 51.6 | 70.7 | 72.3 | 62.05 |
> | 0.5 | 54.0 | 51.3 | 71.6 | 75.1 | 63.00 |
> | 0.7 | 52.9 | 51.1 | 71.6 | 72.2 | 61.95 |
> | 0.9 | 53.0 | 51.0 | 71.3 | 72.1 | 61.85 |
>
>
> **Sensitivity analysis on $\zeta^{\ast}$ with $\mu^\ast = 0.5$**
>
>
> | $\zeta^{\ast}$   | A    | C    | P    | R    | Avg.    |
> |------|------|------|------|------|---------|
> | 0.01 | 52.5 | 51.5 | 71.5 | 72.4 | 61.975  |
> | 0.1  | 54.0 | 51.3 | 71.6 | 75.1 | 63.000  |
> | 0.5  | 53.6 | 52.0 | 71.9 | 72.3 | 62.450  |
> | 1.0  | 54.7 | 51.2 | 72.0 | 73.0 | 62.725  |
>
>
> ---------------------

---

> ### Author Response · Authors · 2025-11-23
> **Response to Reviewer NoZD 2/3**
>
> ## Sensitivity analysis on $T_s$  and $T_f$
>
> Similarly, we fix $T_s$ = 1 and $T_f$ = 1 in our experiments. With $T_s$  = 1 and $T_f$  = 1, we simply ensure that the forecaster is updated as frequently as the student, preventing the forecaster weights becoming stale as the student representation evolves during training. To assess how the frequency of updates to the forecaster affect the downstream OOD performance of the student network, we conduct a sensitivity analysis by varying  $T_s$  and $T_f$ across several configurations shown below. As shown, we observe a small but consistent drop in average OOD performance in these alternative configurations. These observations suggest that the forecaster network is negatively affected in both cases of (1) infrequent updates (10; 1) (lagging behind current student configuration) and too frequent updates (10; 10 / 1; 10) (overfitted to current student configuration).
>
> | ($T_s$, $T_f$) | A    | C    | P    | R    | Avg.    |
> |----------|------|------|------|------|---------|
> | (1; 1)   | 54.0 | 51.3 | 71.6 | 75.1 | 63.000  |
> | (10; 1)  | 53.0 | 51.5 | 71.3 | 73.1 | 62.225  |
> | (1; 10)  | 53.4 | 51.2 | 71.8 | 72.7 | 62.275  |
> | (10; 10) | 52.3 | 52.4 | 71.3 | 71.9 | 61.975  |
>
>
>
> -------------
>
>
> ## Training time Computational Cost for Forecaster
>
> The forecaster network is designed to be extremely lightweight. **In our ResNet18 Vision experiments, the total number of parameters in the forecaster network is 197**, while the **student backbone has 11689512 ≈ 11.7 million parameters (forecaster is at the scale of 0.00168% of the student model)**. The FLOPs and memory contribution stemming from the forecaster are at a negligible scale. **For the Text experiments, the forecaster is ~0.004% of the scale of the student network** (google/bert_uncased_L-4_H-256_A-4)**. Therefore, the training-time overhead from the forecaster is minimal. As the forecaster is discarded at inference, the test-time cost remains identical to ERM and Vanilla KD.

---

> ### Author Response · Authors · 2025-11-25
> **Response to Reviewer NoZD 3/3**
>
> *Insufficient theoretical justification: Why should early layer confidence specifically help with OOD generalization rather than just indicating general sample difficulty? The paper treats these as equivalent without rigorous justification. Early layers capture low-level features—the connection to domain shift is unclear.*
>
> **Domain Generalization Guarantees**
>
> We thank the reviewer for their suggestions on the theoretical implications. We provide a more rigourous theoretical viewpoint in **Supplementary Section B and Section C** as to how early layer wise signals gives a good representation of feature variation across domains and thereby providing an upperbound on the worst domain generalization error. At a high level, different layers of the model captures different representations of the training instance. While in previous works, in expected risk minimization (ERM) setting, only last layer based feature variation across domains is explored, we provide a more grounding substitute to how signals from different layers contribute to the feature variation across domains.
>
> In addition, we also showcase that student trained with forecaster which takes into account the early layer based uncertainty signals like entropy and confidence margin, has a lower generalization error than any vanilla ERM model.
>
> **Convergence guarantees for Student and Forecaster model**
>
> Lastly we also showcase convergence guarantees for both the student model and the forecaster network as part of their interleaved training.
>
>
> -----------------
>
>
> Thank you again for the helpful suggestions and the important feedback which we have incorporated in the revised draft and will further solidify in the final version of the paper. We appreciate your reassurance given your positive inclination towards accepting the paper.
>
> We would appreciate a reconsideration of the current score and an appropriate increase to reflect the extended merits of our work.

---

### Official Review · Reviewer_uDoj · 2025-11-01

**Soundness:** 3
**Presentation:** 3
**Contribution:** 2
**Rating:** 4
**Confidence:** 3

**Summary:**

The paper proposes a student-centric, adaptive KD scheme that learns an instance-wise weight to balance cross-entropy vs. KL terms using a lightweight “forecaster” fed by early-layer readouts (stacked intermediate logits) plus uncertainty signals (entropy and a confidence margin). The forecaster is trained with a correctness-prediction objective and its outputs are stabilized via a batch-standardized sigmoid adjustment before modulating the student loss; training alternates between updating the student/auxiliary heads on train splits and the forecaster on a held-out validation split, and all auxiliaries are discarded at inference. Reported results indicate consistent OOD gains over vanilla KD and DG baselines across multiple benchmarks.

**Strengths:**

1. Student-centric, instance-adaptive weighting grounded in early-layer signals and explicit uncertainty features is a clean, modular idea that avoids hand-crafted rules.

2. No inference-time overhead: the forecaster and auxiliary heads are training-only, preserving deployment efficiency.

3. Clear base loss formalization (CE + KD with temperature) and a simple correctness-based objective for the forecaster make the method easy to implement and analyze.

**Weaknesses:**

1. You call (L^{(S)}*{\text{tot}}) a “convex combination” yet allow (\alpha,\beta\in\mathbb{R}) and only later constrain (\alpha+\beta=1); non-negativity isn’t enforced. Moreover, the KD term is defined with temperature (\tau) but the standard (\tau^2) factor that keeps gradient magnitudes comparable to CE is omitted, and later the forecaster weight replaces ((\alpha,\beta)) as ((1-w*{\text{adj}})) and (w_{\text{adj}}) without any scale calibration. Together, these choices make the optimization ill-conditioned and sensitive to (\tau). Please formalize the constraints (e.g., (\alpha,\beta\ge0,\ \alpha+\beta=1)) and the gradient scaling.

2. The adjusted weight (w_{\text{adj}}=\sigma!\left(\varsigma^*,\frac{z_f-\mu_B}{\sigma_B}+\mu^*\right)) depends on batch mean/STD ((\mu_B,\sigma_B)) and free hyperparameters ((\mu^*,\varsigma^*)). This injects stochastic, batch-size-dependent drift into the loss, offers no invariance guarantees, and couples optimization to mini-batch composition. A principled alternative would derive a calibration mapping from a proper scoring rule or isotonic/logistic calibration fitted on a disjoint set, with stability guarantees. At minimum, clarify how ((\mu^*,\varsigma^*)) are selected and provide conditions ensuring (w_{\text{adj}}\in(0,1)) does not collapse.

3. The forecaster is trained as a binary classifier of student correctness on a “meta-validation” split while the student and auxiliary heads are frozen, then roles are swapped. This alternating scheme implicitly defines a bilevel problem, yet there is no convergence analysis, fixed-point characterization, or even a guarantee that the forecaster’s risk aligns with the downstream student objective under distribution shift between the training and validation partitions. Please either cast the method as bilevel optimization (and justify the alternating updates) or provide a stability argument for Algorithm 1’s cycle.

4. Several mathematical details are unclear: (i) in Eq. (5) (p_K(\cdot)) is described as “probability … classified as label (K)” while (K) also denotes the number of classes—this is ambiguous; (ii) the forecaster loss in Eq. (6) uses inner-product notation (\langle \cdot,\cdot\rangle) without specifying whether this is summed over the minibatch or averaged, and whether class imbalance is handled; (iii) the forecaster’s 1D convolution “over stacked intermediate logits” lacks a precise tensor layout (layer-major vs class-major) and dimensionalities, which matter for reproducibility. Tighten the notation, explicitly define shapes, and disambiguate symbols reused for different roles.

**Questions:**

see the weakness above

---

> ### Author Response · Authors · 2025-11-25
> **Response to Reviewer uDoj 1/2**
>
> We thank the reviewer for their time in reviewing our submission and providing feedback. We’re glad that the core ideas came through clearly, especially the simplicity of the instance-adaptive weighting scheme via early-layer readout signals. We also appreciate the reviewer noting that the approach adds no overhead at inference, which was an important design goal for us. Finally, the comments on the clarity and ease of implementation of the loss setup and the forecaster objective are encouraging, as we aimed to keep the method lightweight and straightforward to reproduce.
>
>
> We hereby address the questions raised by the reviewer
>
> -----------------
>
>   *You call $L^{(S)}_{\text{tot}}$ a “convex combination,” yet allow $\alpha$, $\beta \in \mathbb{R}$ and only later impose the constraint  $\alpha + \beta = 1$; non-negativity is not enforced.*
>
> We thank the reviewer for raising this question. In our formulation, it is implied that $\alpha$ and $\beta$ are both non-negative due to the fact that we consider them as coefficients over two losses: (1) the ground-truth Cross Entropy Loss and (2) the teacher’s KD Loss. Additionally, in our proposed methodology, we replace $\alpha$ and $\beta$ with  (1 - $w_{adj}$) and $w_{adj}$, respectively, which is obtained using a sigmoid transform on the forecaster logits, thereby being inherently non-negative. We have incorporated the change in our current updated version (marked in red).
>
> -----
>
> *Moreover, the KD term is defined with temperature $\tau$ but the standard $\tau^2$ factor that keeps gradient magnitudes comparable to CE is omitted, and later the forecaster weight replaces $(\alpha,\beta)$ as $((1-w{\text{adj}}))$ and $(w_{\text{adj}})$ without any scale calibration. Together, these choices make the optimization ill-conditioned and sensitive to ($\tau$).*
>
> The omission of the standard $\tau^2$ factor in the KD term was an oversight in the original formulation. We have included that in the formulation in line 135-136. In all our experiments setting we consider, $\tau$ fixed across baselines.
>
> ------
>
>
> *The forecaster is trained as a binary classifier of student correctness on a “meta-validation” split while the student and auxiliary heads are frozen, then roles are swapped. This alternating scheme implicitly defines a bilevel problem, yet there is no convergence analysis, fixed-point characterization, or even a guarantee that the forecaster’s risk aligns with the downstream student objective under distribution shift between the training and validation partitions. Please either cast the method as bilevel optimization (and justify the alternating updates) or provide a stability argument for Algorithm 1’s cycle.*
>
> We appreciate the reviewer for their observation. We include in, Section 5, the same problem posed as a bilevel objective optimization problem and associated convergence steps in Supplementary section Section B.2. In the interleaved schedule, the post-hoc adjustment on the forecaster’s logits allows the notion of a sample’s difficulty to be relative to what the student currently sees at a particular training step, and ensures that the student continues to receive supervision from both teacher and ground-truth even late in the distillation process. Since, $\mu^\ast$ is chosen equal to $\beta$, the forecaster's predictions are centered around $\beta$ from step 0 of training. This ensures that the optimization is stable even before the forecaster has learned meaningful information.

---

> ### Author Response · Authors · 2025-11-25
> **Response to Reviewer uDoj 2/2**
>
> *The adjusted weight $w_{\text{adj}}=\sigma(\varsigma^\ast(\frac{z_f-\mu_B}{\sigma_B})+\mu^\ast)$ depends on batch mean/STD $(\mu_B,\sigma_B)$ and free hyperparameters $(\mu^\ast,\varsigma^\ast)$. This injects stochastic, batch-size-dependent drift into the loss, offers no invariance guarantees, and couples optimization to mini-batch composition. A principled alternative would derive a calibration mapping from a proper scoring rule or isotonic/logistic calibration fitted on a disjoint set, with stability guarantees. At minimum, clarify how $(\mu^\ast,\varsigma^\ast)$ are selected and provide conditions ensuring $w_{\text{adj}}\in(0,1)$ does not collapse.*
>
> The primary purpose of selecting $\mu^\ast$ is to keep the forecaster weights centered around a fixed point throughout the student’s training run. This allows the forecaster weights to never collapse to extreme values and cause the training signal to come solely from ground-truth or soft-labels. For our experiments $\mu^\ast$ is kept the same as $\beta$ for a fair comparison to vanilla KD. Specifically, we keep $\beta = \mu^\ast = 0.5$ as pragmatic defaults (equal contribution from soft and hard supervision signals) without requiring dataset or domain-specific tuning. Based on reviewer suggestions, we conduct sensitivity analysis on the choices of $\mu^\ast$ and $\varsigma^\ast$ on OfficeHome for various configurations of $\mu^\ast$ and $\varsigma^\ast$.
>
>
> **Sensitivity analysis on $\mu^{\ast}$ with $\zeta^{\ast}$=0.1**
>
>
> | $\mu^{\ast}$  | A    | C    | P    | R    | Avg.  |
> |-----|------|------|------|------|-------|
> | 0.1 | 52.2 | 51.8 | 69.9 | 71.9 | 61.45 |
> | 0.3 | 53.6 | 51.6 | 70.7 | 72.3 | 62.05 |
> | 0.5 | 54.0 | 51.3 | 71.6 | 75.1 | 63.00 |
> | 0.7 | 52.9 | 51.1 | 71.6 | 72.2 | 61.95 |
> | 0.9 | 53.0 | 51.0 | 71.3 | 72.1 | 61.85 |
>
>
> **Sensitivity analysis on $\zeta^{\ast}$ with $\mu^\ast = 0.5$**
>
>
> | $\zeta^{\ast}$   | A    | C    | P    | R    | Avg.    |
> |------|------|------|------|------|---------|
> | 0.01 | 52.5 | 51.5 | 71.5 | 72.4 | 61.975  |
> | 0.1  | 54.0 | 51.3 | 71.6 | 75.1 | 63.000  |
> | 0.5  | 53.6 | 52.0 | 71.9 | 72.3 | 62.450  |
> | 1.0  | 54.7 | 51.2 | 72.0 | 73.0 | 62.725  |
>
>
> ---------------------
>
>
>
>
> Given that the configurations $\mu^\ast$ and $\varsigma^\ast$ do not degrade OOD performance sharply, we note that the OOD performance gain depends mostly on the pre-adjustment weights based on the early layer logits and uncertainty signals: confidence margin, prediction entropy. Towards the question on batch size dependence, it is important to note that the training loss (gradient) itself is calculated over a minibatch making it a batch-dependent process by default. Hence, all operations are performed in a batch dependent manner. The batch-based normalization is a widely accepted training standard. It allows the notion of a sample’s difficulty to be relative to what the student currently sees at a particular training step, and ensures that the student continues to receive supervision from both teacher and ground-truth even late in the distillation process.
>
>
> **New Theoretical Guarantees** : In addition to above experiments, we provide new theoretical insights and results in Supplementary(Appendix) Section B, Subsection B.2 and Subsection C.2 (on convergence analysis of forecaster network)
>
>
> -----------
>
>
> *Several mathematical details are unclear: (i) in Eq. (5) ($p_K(\cdot)$) is described as “probability … classified as label (K)” while (K) also denotes the number of classes—this is ambiguous; (ii) the forecaster loss in Eq. (6) uses inner-product notation ($\langle \cdot,\cdot\rangle$) without specifying whether this is summed over the minibatch or averaged, and whether class imbalance is handled; (iii) the forecaster’s 1D convolution “over stacked intermediate logits” lacks a precise tensor layout (layer-major vs class-major) and dimensionalities, which matter for reproducibility. Tighten the notation, explicitly define shapes, and disambiguate symbols reused for different roles.*
>
> We thank the reviewer for their suggestion to make the notations clearer. Keeping this in consideration, we make the following changes for Eq 5 and Eq 6 in red. We have also included all domain shapes and vector shapes and fixed any loose notations wherever deemed necessary.
>
> Kindly check line nos. Main Paper (289-290), (252-253), (312) , and Appendix line nos. (897-903) for  logits stacking formulation (marked in red).
>
> --------
>
>
> ---
>
>
> Thank you again for the helpful suggestions and valuable feedback, which we have incorporated in the revised draft, and further solidify in the final revision of the paper. We hope that our responses have adequately addressed your concerns.
>
> We kindly request your support in reflecting this through an appropriate score increase to help strengthen the case for acceptance.

---

### Author Response · Authors · 2025-11-26
**Overall response and revision summary**

We thank all the reviewers for their thoughtful and constructive feedback. In response, we have substantially improved the clarity of the manuscript, added new experiments, and provided additional theoretical justification and insights. We have addressed each concern raised, with detailed point-by-point responses posted as comments on OpenReview highlighting the key revisions and clarifications. All new or updated content in the revised manuscript is marked in red.

---

### Author Response · Authors · 2025-12-03
**Rebuttal Summary (1/2)**

As the discussion phase comes to an end, we would like to sincerely thank all the reviewers for their constructive feedback on our submission. We appreciate the suggestions and questions put forth by the reviewers that have helped us improve our submission with better theoretical grounding, additional experiments over the discussion period **(Changes marked in red in subsequent revisions of the paper)**.

During the discussion period, we were able to successfully address all major concerns raised by the reviewers. We are grateful to Reviewer **XXf7** for their active engagement during the discussion phase. Following our clarifications on theoretical contribution in our proposed method and extensive additional experiments, **Reviewer XXf7 raised their score**. This change occurred before the news of OR leak and we believe it was a genuine acknowledgement of our rebuttal responses. Although other reviewers were unable to participate due to the shortened discussion phase, we believe that we have effectively addressed all questions raised by them.

**Summary of Strengths as Identified by the Reviewers**

1. **Student-centric, adaptive design**: The proposed method focuses on when and how the student should trust the teacher, leveraging early-layer uncertainty signals for per-sample weighting via a meta-model rather than relying on hand-crafted rules, a direction that has been relatively underexplored **(uDoj, JkxR, XXf7)**.

2. **No inference-time overhead**: Auxiliary heads and Forecaster meta-network are discarded after training, ensuring inference-time efficiency is preserved **(uDoj, NoZD, JkxR)**.

3. **Clear motivation and pragmatic aim**: The proposed method builds on the observation that vanilla KD already provides strong OOD gains and addresses per-sample variability and teacher bias **(JkxR, NoZD)**.

4. **Strong empirical validation**: Approach demonstrates consistent improvements across four benchmarks (OfficeHome, PACS, VLCS, TerraIncognita) with both ResNet152 and ViT-L/16 teachers **(NoZD)**.

5. **Transparent experimental setup**: Experimental setup built upon DomainBed, follows established protocol, and compares against multiple baselines including best-performing DG algorithms per dataset **(NoZD)**.

6. **Thoughtful design choices**: Method combines CE + KD with per-sample weighting with a simple correctness-based objective for the forecaster; post-hoc adjustment mechanism to forecaster is well-motivated and ablations highlight importance of usage uncertainty features as input to forecaster meta-network **(uDoj, NoZD)**.

**Reviewer Concerns, Key Revisions and New Results**

1. **Sensitivity analysis on newly-introduced hyperparameters (Re: uDoj, JkxR, XXf7)**: We presented the pragmatic rationale behind the chosen hyperparameter values in our proposed algorithm. Based on reviewer suggestions, we conducted an extensive sensitivity analysis on all newly-introduced hyperparameters. Our analysis shows that OOD performance does not degrade sharply across various choices of these hyperparameters, indicating that the method exhibits low, explainable sensitivity to these hyperparameters.

2. **Theoretical and Technical Clarifications (Re: uDoj, NoZD, JkxR, XXf7)**: We clarified the theoretical grounding of our work, in particular, we attached theoretical justification to each design choice (using early student layers; alternating training; applying post-hoc adjustment) under OOD conditions in Appendix B and C, in the revised draft of the paper. Additionally, we addressed several implementation-specific concerns (training overhead, forecaster design, interleaved schedule) during the discussion phase.

3. **Generalizability and Scalability (Re: NoZD, JkxR, XXf7)**: In addition to the benchmarks presented in the paper, we conducted extensive experiments showcasing the generalizability of our work. First, we presented results on two NLP ID-OOD pairs of datasets using transformer-based BERT, where our proposed method outperforms vanilla KD, showcasing effectiveness of our approach on a new modality and a transformer-based architecture. Second, we presented DG results on ColoredMNIST, which is a representative correlation-shift OOD benchmark, showcasing robustness of our proposed method on spurious correlation based OOD scenario. For Scalability, we presented positive DG results on OfficeHome using a ResNet50 student. We believe that these additional experiments highlight wide applicability of our proposed adaptive KD approach.

4. **Additional Baselines (Re: JkxR, XXf7)**: We clarified why the suggested baselines are orthogonal to our setting. These methods assume extensive hyperparameter search (DiWA), multi-modal teacher supervision (RISE) or multi-expert architectures with online KD (BOLD), which differ fundamentally from our student-centric design. Adding these baselines would conflate distinct problem formulations. However, we acknowledge these approaches as potential add-on alongside our method.

---

> ### Author Response · Authors · 2025-12-03
> **Rebuttal Summary (2/2)**
>
> **Conclusion**
>
> The reviewers have collectively highlighted the significance of this work. In this work, we present a relatively underexplored student-centric adaptive knowledge distillation design to address out-of-domain generalization in small networks suitable for real-world deployment. We propose an adaptive per-sample weighting scheme to balance Cross Entropy vs. KD loss in an offline distillation process using a lightweight meta-network *forecaster* fed by early-layer readouts (stacked intermediate logits from student's early layers) plus uncertainty signals (entropy and a confidence margin of early layer prediction distribution). Forecaster, and early layer prediction heads are discarded at inference-time, providing zero additional overhead at inference, as compared to vanilla KD. Our proposed approach, showcases consistent OOD performance improvements over 6 benchmark datasets (OfficeHome, PACS, VLCS, TerraIncognita, ColoredMNIST, MPRC-QQP, MNLI-MNLImismatched) against vanilla KD, highlighting wide applicability of our proposed adaptive KD approach.
>
> We are grateful to the Reviewers and the AC for their time to review our work. Based on reviewer suggestions, the additional experiments and theoretical analysis have significantly strengthened the paper. We hope that our work will provide valuable insights to the research community.
>
> Best Regards,
>
> The Authors

---

### Note · Authors · 2026-01-27

I have read and agree with the venue's withdrawal policy on behalf of myself and my co-authors.

---

### Meta-Review · Area_Chair_3FmG · 2025-12-19

**Summary:**

The initial scores are 2x 2 (reject), 1x 4 (borderline reject), and 1x 6 (borderline accept).

Specifically, reviewer XXf7, who voted for reject, criticised the lack of experiments on large datasets like DomainNet and pointed out that the ablation study is insufficient.

Reviewer JkxR also voted for reject. This reviewer found the method incremental and improvement limited. The reviewer also pointed out several issues including insufficient experiments and benchmarks (missing DomainNet), missing baselines (more recent KD methods), lack of analysis on hyperparameter sensitivity, and lack of justification on why early-layer confidence helps OOD generalization.

Reviewer NoZD gave the rating of 6. Similar to XXf7 and JkxR, NoZD also found the improvement limited and the experiments insufficient for justifying the contributions (need larger scale datasets).

Reviewer uDoj gave the rating of 4. This reviewer's questions are mainly technical: sensitivity of hyperparameter, no convergence analysis, and unclear math formulations.

Key concerns related to experiment with larger datasets and limited improvement remain unaddressed during rebuttal.

**Reviewer Concerns:**

The rebuttal provided additional results about hyperparameter sensitivity and generality of the method to different architectures. The rebuttal also clarified some issues related to the technical designs. These may have addressed some of the concerns from the reviewers. However, key concerns including lack of experiments on larger datasets like DomainNet and limited improvement of the approach remain unaddressed. The rebuttal attributed these problems primarily to a lack of computing resources. From the AC's own experience, the missing experiments mentioned by the reviewers could well be run by consumer-grade GPUs. To fully justify the contributions, the AC suggests that the paper expand the experiments to include larger scale datasets like DomainNet, as well as more architectures like ViT in vision tasks.

**Reviewer Scores:**

After carefully examining the paper, reviews, and rebuttal, the AC predicts that reviewer uDoj may raise the score from 4 to 6 or 8 while other scores would be remained the same. So the final predicted scores are 2, 2, 6, 6/8. Since the reviewers are more negative than positive and there are lingering concerns, the AC recommends rejection.

---

### Decision · Program_Chairs · 2026-01-26

Reject